# Leveraging data-driven self-consistency for high-fidelity gene expression recovery

Md Tauhidul Islam [1], Jen-Yeu Wang[1], Hongyi Ren [2], Xiaomeng Li[1], Masoud Badiei Khuzani[1], Shengtian Sang[1], Lequan Yu[1], Liyue Shen[2], Wei Zhao[1] & Lei Xing [1] ✉

Single cell RNA sequencing is a promising technique to determine the states of individual cells and classify novel cell subtypes. In current sequence data analysis, however, genes with low expressions are omitted, which leads to inaccurate gene counts and hinders downstream analysis. Recovering these omitted expression values presents a challenge because of the large size of the data. Here, we introduce a data-driven gene expression recovery framework, referred to as self-consistent expression recovery machine (SERM), to impute the missing expressions. Using a neural network, the technique first learns the underlying data distribution from a subset of the noisy data. It then recovers the overall expression data by imposing a self-consistency on the expression matrix, thus ensuring that the expression levels are similarly distributed in different parts of the matrix. We show that SERM improves the accuracy of gene imputation with orders of magnitude enhancement in computational efficiency in comparison to the state-of-the-art imputation techniques.

Single-cell RNA sequencing (scRNA-seq) has emerged as an effective tool for a variety of cellular analysis tasks such as quantifying the state of individual cells[1,2], identifying novel cell subtypes[3,4], assessing progressive gene expression (cell trajectory analysis)[5–7], performing spatial mapping[8,9] and finding differentially expressed genes[10,11]. Despite its prevalence in computational biology, due to noise and low transcript capture efficiency, the resulting gene expression matrix from scRNA-seq is typically sparse, which often results in a loss of important biological information[12–14]. In the past decade, intense research has been devoted to address the computational challenges in recovering omitted gene expressions. The developed techniques can be broadly divided into three categories[15]. In the first category, the sparsity of expression data is modeled using probabilistic theories. SAVER[12], SAVER-X[16], bayNorm[17], scImpute[13], and VIPER[18] are the prominent methods of this group. The second category includes the imputation methods that utilize averaging/smoothing to recover the expression values. DrImpute[19], MAGIC[14], and k-NN smoothing[20] are the most popular techniques of this category. The last category is based on the reconstruction of data either using deep learning (AutoImpute[21], DCA[22], DeepImpute[23], SAUCIE[24], scScope[25], scVI[26]) or low-rank matrix

assumption (mcImpute[27], PBLR[28]). Overall, the model-based techniques assume a specific model for the expression data, which may limit their applicability in some practical cases. For example, SAVER extracts information from correlated genes and employs a penalized regression model to impute the data. It assumes that the gene expressions in a cell follow a gamma-Poisson distribution, which may not be appropriate in many cells, including those with low gene expression values[12]. The smoothing-based methods extract information from similar cells to impute the expression data. However, the averaging effect in these methods may potentially eliminate variability in gene expressions across cells[14]. The deep learning-based methods learn from the data to perform imputation without imposing any assumption. However, as the deep learning techniques are purely data-driven, the data imputation process is essentially a complex black-box operation with limited transparency. Low-rank matrix-based techniques, such as mcImpute, use a matrix completion approach to impute the data with considerations of both gene-gene and cell-cell relationships. However, in mcImpute, all zero expression values are treated as dropout events, which may lead to spurious results in some practical scenarios. Generally, the analytical techniques (model-based,

[1]Department of Radiation Oncology, Stanford University, Stanford, CA 94305, USA. [2]Department of Electrical Engineering, Stanford University, Stanford, CA 94305, USA. ✉e-mail: lei@stanford.edu

smoothing-based, and low-rank matrix-based methods) are not scalable to large datasets since the entire dataset must be processed as a whole.

Here we propose a novel strategy, self-consistent expression recovery machine (SERM), to recover the missing gene expression values by enforcing data self-consistency, a unique characteristic of high-dimensional datasets. In general, the self-consistency in datasets implies that if two sets of data points arise from similar sources (such

as gene counts in similar cells), they should have similar distributions[29–31]. Computationally, SERM first learns the underlying data distribution (pattern of self-consistency) using deep learning and then imputes the expression values by ensuring the self-consistency of the data. The approach leverages a deep learning optimization to extract the latent representation of the data and reconstruct the denoised expression values (Fig. 1-step 1). It then uses a curve fitting technique to learn a probability distribution from a parametric family

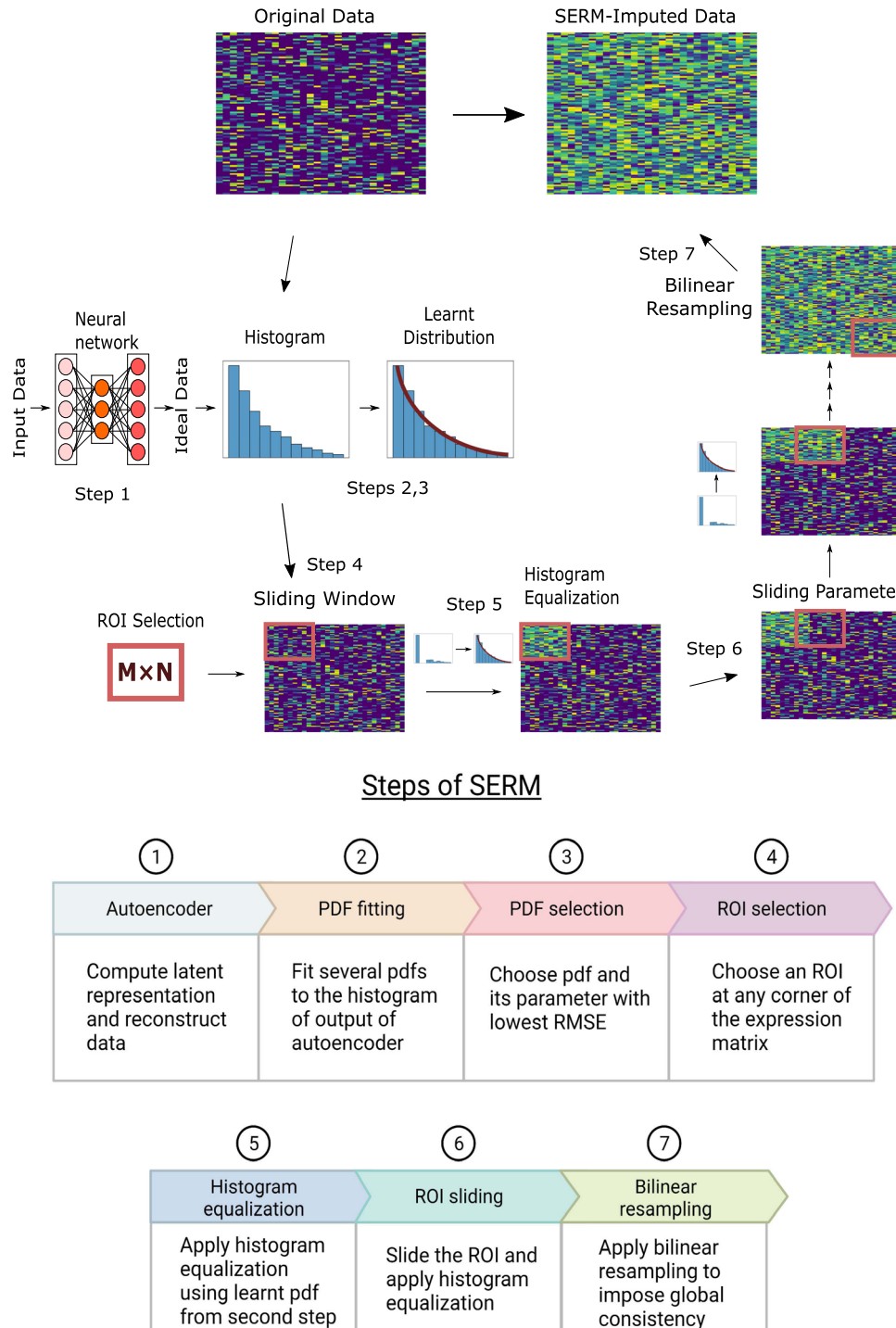

**Fig. 1 | Workflow of SERM.** A subset of the expression matrix is inputted into an autoencoder network. To learn the distribution function that best describes the reconstructed data by the autoencoder, different pdfs are fitted to the histogram of the reconstructed data. Next, an ROI is selected, and histogram equalization is performed on that ROI using the learned pdf in the previous step. The ROI then slides along the x and y direction throughout the expression matrix, and histogram equalization is performed on each ROI. All the regions are then interpolated using bilinear resampling in the final step to impose global consistency.

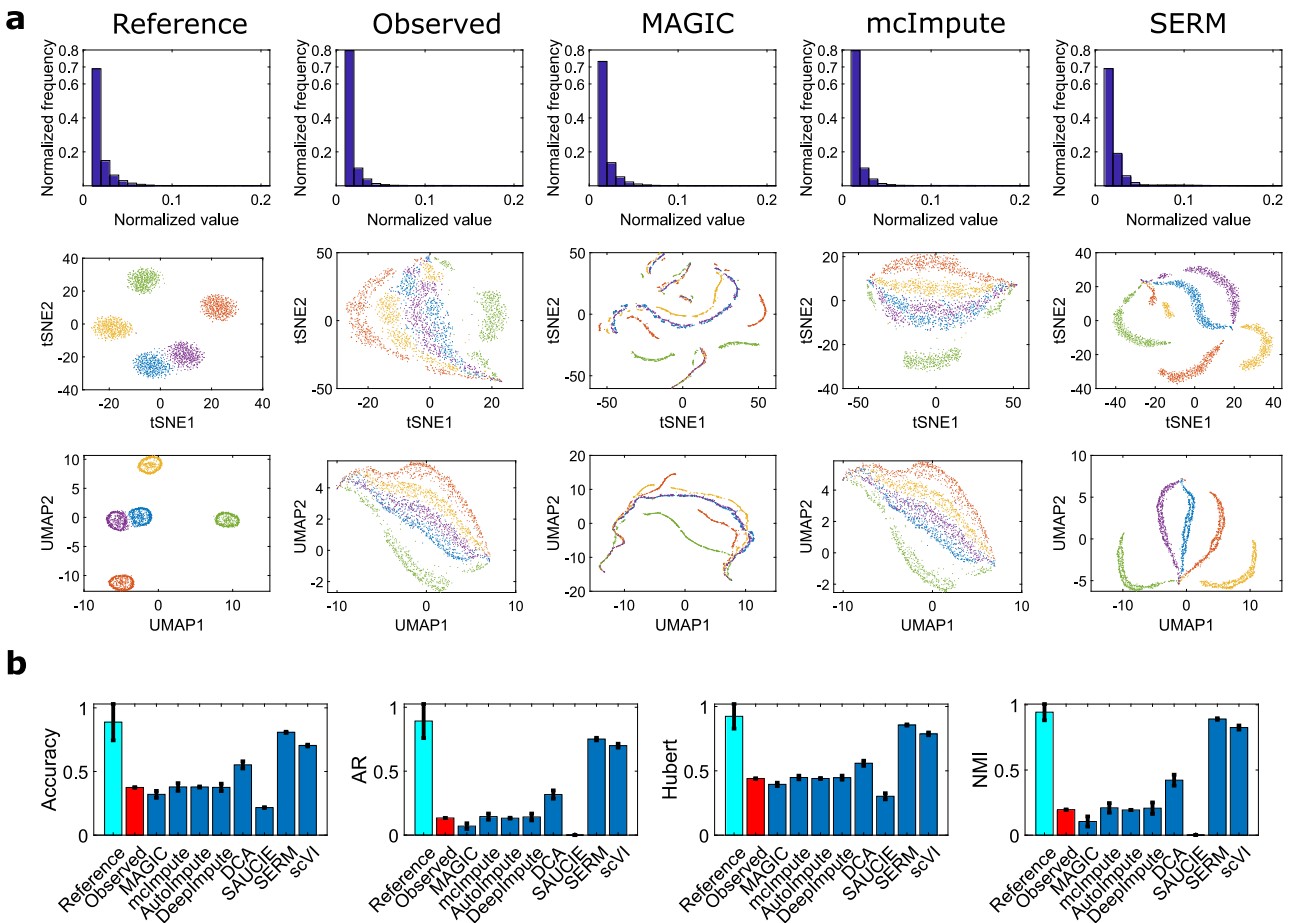

**Fig. 2 | Analysis of simulated scRNA-seq data with five classes.** The histograms of the reference data, observed data (1% sampling efficiency), and imputed data by MAGIC, mcImpute, and SERM are shown in the first row of (**a**). Visualization of reference, observed, and imputed data by t-SNE and UMAP are shown in the second and third rows, respectively. t-SNE and UMAP results from SERM imputed data are much better in separating the classes, whereas MAGIC degrades the data due to imputation. The clustering accuracy and cluster quality indices for UMAP visualizations of imputed data from different methods are shown in (**b**). Data are presented as mean values +/− standard deviation (SD). Error bars represent the standard deviation of the indices for $n = 1000$ different initializations of k-means clustering. Source data are provided as a Source Data file.

to represent the reconstructed expression values from the optimization (Fig. 1-steps 2, 3). Next, a rectangular region of interest (ROI) is selected within the expression matrix (Fig. 1-step 4) and imputation is performed based on histogram equalization using the learned probability distribution (Fig. 1-step 5). The imputation calculation proceeds progressively, with the ROI shifting along the x- and y-axes of the gene expression matrix for each window position. This process can be regarded as a 'sliding window' approach (Fig. 1-step 6). In the last step, all the ROIs are interpolated using bilinear resampling to achieve the global consistency of the expression values (Fig. 1-step 7, supplementary Fig. S1). Because of this divide-and-conquer strategy, SERM is scalable to large datasets and can impute the datasets much faster than other analytical or deep learning-based methods.

We showcase the efficacy of SERM on various benchmark datasets and demonstrate its superior performance over the existing baseline methods. Compared to alternative techniques, SERM consistently achieves significantly better imputation accuracy and speed. The method promises to substantially improve the way that scRNA-seq data is processed and utilized in biomedical research.

## Results

In this section, we first present the imputation results for synthetic and benchmark gene sequence datasets and show the benefits of SERM in visualization and clustering applications. In these studies, the performance of different imputation methods (MAGIC, mcImpute, AutoImpute, DeepImpute, SAUCIE, scVI, DCA, and SERM) are measured in terms of (1) visualization quality, (2) correlation of the imputed data with the reference data, (3) clustering accuracy, (4) normalized mutual information (NMI)[32], and (5) two cluster quality indices (adjusted Rand (AR)[33], and Hubert[34]). We then show the superior performance of SERM in analyses of the cell trajectory datasets. Here the performance of different methods is assessed by using (1) trajectory quality, (2) correlation of the imputed data with original data, and (3) correlation of computed pseudotime from the imputed data with the ground truth time points. At the end, the capability of SERM in imputing large genomic datasets (the human cell landscape and the mouse cell atlas) is demonstrated to show the strong computational ability of SERM.

### SERM provides better data recovery and yields improved high dimensional data visualization and clustering

We first use the Splatter simulator to generate simulated data of 5 classes without dropout[35], where each class consists of 500 cells and 1200 genes (see "Methods" for simulation parameters). The dropout-free data are referred to as the 'reference data'. We sampled the reference data following ref. 12 (see "Methods") at 1% efficiency to simulate the dropout-affected data (referred to as 'observed data'). The histograms of the reference and observed data are shown in the first and second columns of the first row of Fig. 2a. As a result of the efficiency loss, the number of zero values in the distribution increases

for the observed data in comparison to the reference data. The histograms of the imputed data from MAGIC, mcImpute, and SERM are shown in columns 3–5. It is seen that both MAGIC and SERM are able to reduce the number of zeros, whereas mcImpute fails to do so. SERM imputes the zero elements of the data with high accuracy, which is reflected by the similar number of zeros in the reference and SERM-imputed data. The superior performance of SERM is also apparent in the t-SNE and UMAP visualizations of the data imputed by using different methods, as shown in the second and third rows of Fig. 2a. Due to the dropout effect, t-SNE and UMAP visualizations of the observed data exhibit classes separated only by small distances with many inaccurately clustered data points. McImpute result is similar to the observed data. Some distortions are observed in the visualization of MAGIC-imputed data. On the other hand, SERM successfully imputes the data with high-quality visualization (Fig. 2 (5th column)). The clustering accuracy and cluster quality indices for all the methods are shown in (b). It is seen that SERM achieves the best result in terms of these indices. scVI performs better than other techniques except SERM.

In the next study, we use data from four different scRNA-seq experiments (cellular taxonomy of the bone marrow stroma in Homeostasis and leukemia[36], classification of cells from mammalian brains[37], data from mouse intestinal epithelium[38] and comparison of engineered 3D neural tissues[39]). Descriptions of the data can be found in the "Methods" section. In our analysis, the raw data are filtered by choosing cells and genes with high expression values to create reference datasets (see "Methods"). The reference datasets are then sampled at different efficiency following ref. 12 to create observed datasets and different methods are used to impute the expression values. We show UMAP visualization of all four datasets imputed by different techniques in Fig. 3 and Supplementary Fig. S2. The UMAP visualizations of the reference data are shown in the first column of Fig. 3. Again, the data classes are better separated in SERM-imputed data as shown in Fig. 3 (cluster improvement indicated by arrow). Some distortions are seen in the data visualizations in most cases for MAGIC- and SAUCIE-imputed data.

Quantitative assessments of different methods are shown in Fig. 4, and Supplementary Figs. S3 and S4. For the case of 5% sampling efficiency, SERM provides a Pearson coefficient better than 0.9 for most of the genes, which is much higher than the results (≤0.4) from most other methods. For the case of 0.1% downsampling, SERM recovers the original expression values with Pearson coefficients of 0.15–0.65, which are much better than the results from other methods. The quantitative evaluation of clustering of the data imputed by different techniques are presented in supplementary Figs. S5–S8, where it is seen that SERM leads to the best cluster quality indices (accuracy, NMI, AR, and Hubert).

## SERM offers reliable cell trajectory analysis

The following studies demonstrate the quality and accuracy of cell trajectory analysis enabled by SERM. For example, the recently developed dimensionality reduction method PHATE[40] can accurately infer cell trajectories from scRNA-seq data. However, the performance of this method (and other trajectory inference methods) can be degraded in the presence of technical noise, such as dropouts. SERM mitigates the effect of dropouts and facilitates inference methods such as PHATE to compute cell trajectories with high fidelity.

The first dataset we consider is acquired by profiling 38,731 cells from 694 embryos across 12 stages of early zebrafish development[2,41]. In our experiments, the raw data is filtered by choosing cells and genes with high expression values to create a reference dataset (see "Methods"). The reference data is then sampled at different efficiencies to create observed datasets. PHATE visualizations of the reference data, observed data (sampled at 0.1% efficiency), and the data imputed by different methods are shown in Fig. 5a and Supplementary Fig. S9. It is

observed that SERM-imputed data yield the best results. The fine details, such as two paths (indicated by arrows) of transitioning of 8 hpf (hours post fertilization) to 10 and 12 hpf is much more obvious in SERM results than the observed dataset. Results from the data imputed by most other methods are somewhat distorted, making it challenging to find the cell trajectories. We note that scVI and DCA perform better than SAUCIE and MAGIC in this case.

The second dataset is scRNA-seq data of human Embryonic Stem (ES) cells differentiated to embryoid bodies (EBs)[40,42]. The PHATE embedding of the EB data, reference, observed, and imputed by different methods, are shown in Fig. 5b and supplementary Fig. S9 (a2-e2). Like the last study, most methods (MAGIC, mcImpute, AutoImpute) yield somewhat distorted results. PHATE visualizations from SERM-imputed data show less distortion, and the trajectories can be seen clearly. Specially, the two paths (indicated by arrows) of transitioning from time point 2 to 4 and 5 are much clearer in SERM-results compared to the observed data.

To quantify the efficacy of the imputation methods in finding the cell trajectories, the observed data and imputed data from different methods are analyzed using monocle[7]. The Pearson correlation coefficients between the ground truth labels and computed pseudotimes by monocle are shown in Fig. 5c. It is seen that for zebrafish development dataset, SERM produces consistently higher Pearson correlation values for all sampling efficiencies. We note that mcImpute, DeepImpute, and scVI provide comparable results to SERM for a few cases. The observed data and imputed data from different methods were also analyzed using TSCAN[43] and Slingshot[44], and the results are presented in supplementary Figs. S41 and S42. The Pearson coefficients between the expression values of the reference and imputed data by different methods are shown in Fig. 6, and the percent improvements in Pearson coefficients are shown in supplementary Fig. S10 for the above two datasets. SERM achieves the highest coefficients in all cases. In Fig. 6 (and Fig. 4), it is seen that as the sampling efficiency increases (reduction of dropout), the performance of most methods improves. We noticed that the performance of DeepImpute is, in some cases, better than other methods in the first two sampling efficiencies (0.1 and 0.2%) with comparable results to SERM. Overall, Magic and SAUCIE are less accurate in genomic data imputation compared to other techniques.

## SERM is capable of imputing large datasets that are computationally prohibitive to many existing techniques

The goal here is to demonstrate that SERM can be used to impute datasets of very large size and complexity. To this end, we analyze two large-scale datasets: the human cell landscape (HCL) and the mouse cell atlas (MCA). These two datasets have about 16,403 million data elements (599,926 cells × 27,341 genes) and 11,665 million data elements (333,778 cells × 34,947 genes), respectively. The HCL dataset has 63 unique cell types and 59 unique tissue types, whereas the MCA dataset has 52 unique cell types and 47 unique tissue types. We found that none of the traditional analytical methods could impute these datasets within a week on a personal computer (PC) with an Intel Core i9 processor and 64GB RAM. The deep learning methods (SAUCIE, DeepImpute, scVI, and DCA) can analyze the datasets within a reasonable time.

To benchmark the performance of SERM for these datasets, we subsampled the raw data by 20-times randomly and analyzed the subsampled data using different methods (computational time required for various approaches is added in supplementary Fig. S40). The results are shown in Fig. 7 for MCA and Fig. 8 for HCL datasets, respectively. For raw MCA data, it is seen that the data classes are clustered together and hard to separate. Similar scenarios also occur for the imputed data for all the methods except SERM and scVI. The data classes are better separated than the raw data in scVI calculations, but the best separation of the data classes is seen in the SERM results.

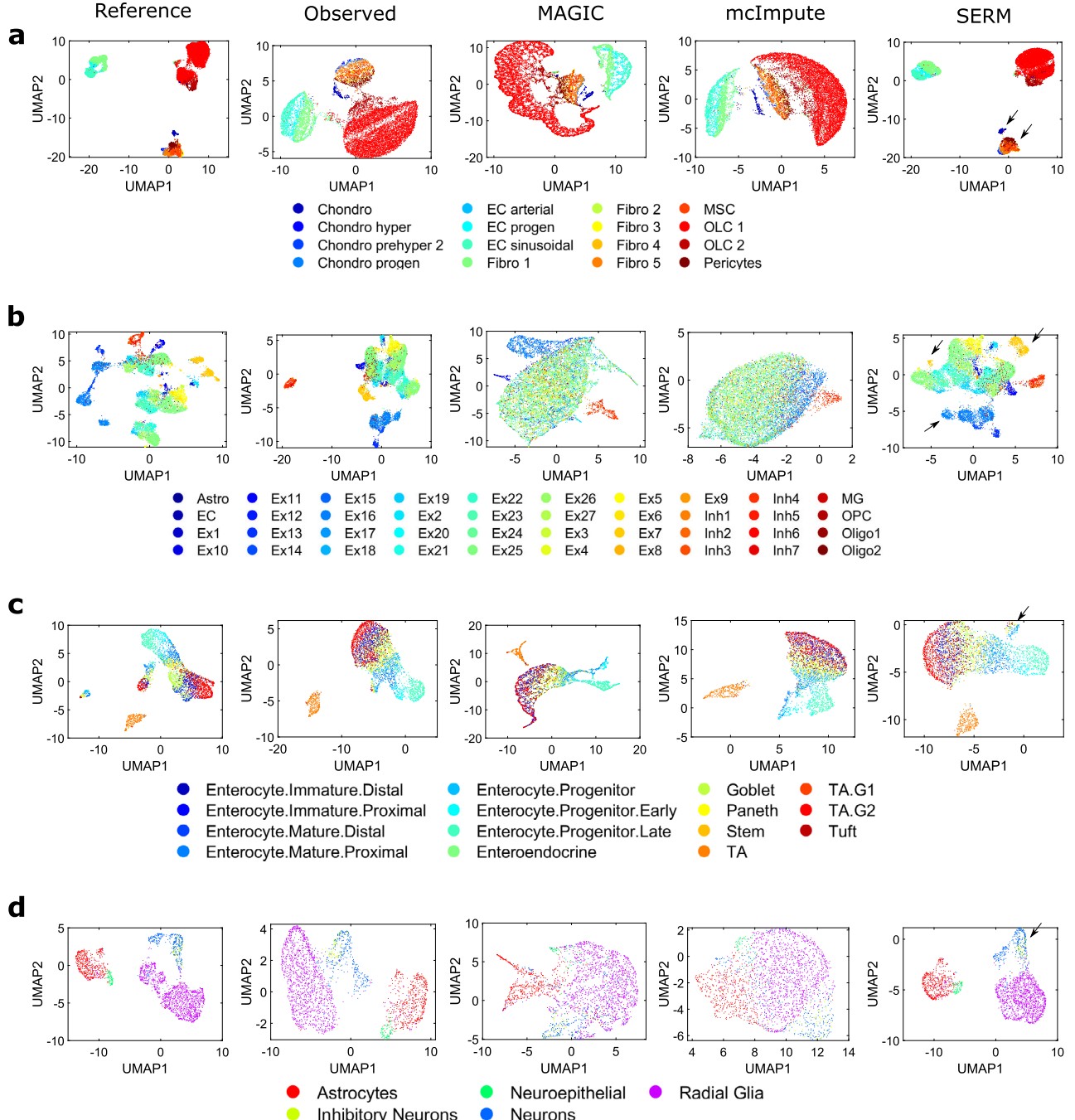

**Fig. 3 | Low dimensional visualization of imputed data from various methods.** UMAP results of the reference data, observed data, imputed data from MAGIC, mcImpute, and SERM for **a** cellular taxonomy, **b** mammalian brain, **c** mouse intestinal epithelium, and **d** 3D neural tissue data. Cellular taxonomy data was sampled at 10% efficiency, and the other three datasets were sampled at 0.1% efficiency. All the classes are better visualized in the SERM imputation. MAGIC and mcImpute distort the data in many cases, whereas SERM retains the consistency of the data intact in all cases. Source data are provided as a Source Data file.

DCA and DeepImpute perform relatively better than SAUCIE and MAGIC. Similar conclusions can be drawn from the results for the HCL data as shown in Fig. 8. The clustering accuracy and three other cluster quality indices (NMI, AR, and Hubert) for different methods are computed and shown in Fig. 9. It is seen that only scVI and SERM can improve the cluster quality after imputations. SERM improves all the indices by at least 15% for MCA data compared to the unimputed data.

Results of the full MCA and HCL datasets for SERM, SAUCIE, and DeepImpute are shown in Supplementary Figs. S35–S39, where it is seen that SERM provides much better qualitative and quantitative performance than other methods. scVI and DCA results are very large (>100 GB) for these datasets, and downstream analyses (t-SNE visualization and computation of cluster quality indices) are not possible to perform because of the limitation of the computer memory.

More experiments are performed on scRNA-seq datasets from different established databases such as IDH-mutant gliomas data (Figs. S11, S12), single-cell data of pediatric midline gliomas (Figs. S13, S14), melanoma intra-tumor heterogeneity data (Figs. S15, S16), Div-Seq data (Figs. S17, S18), intestinal immune cell atlas (Figs. S19–S21), Tabula Muris (TM) dataset (section 13) and the results are included in the supplementary. From these studies, we observed that the

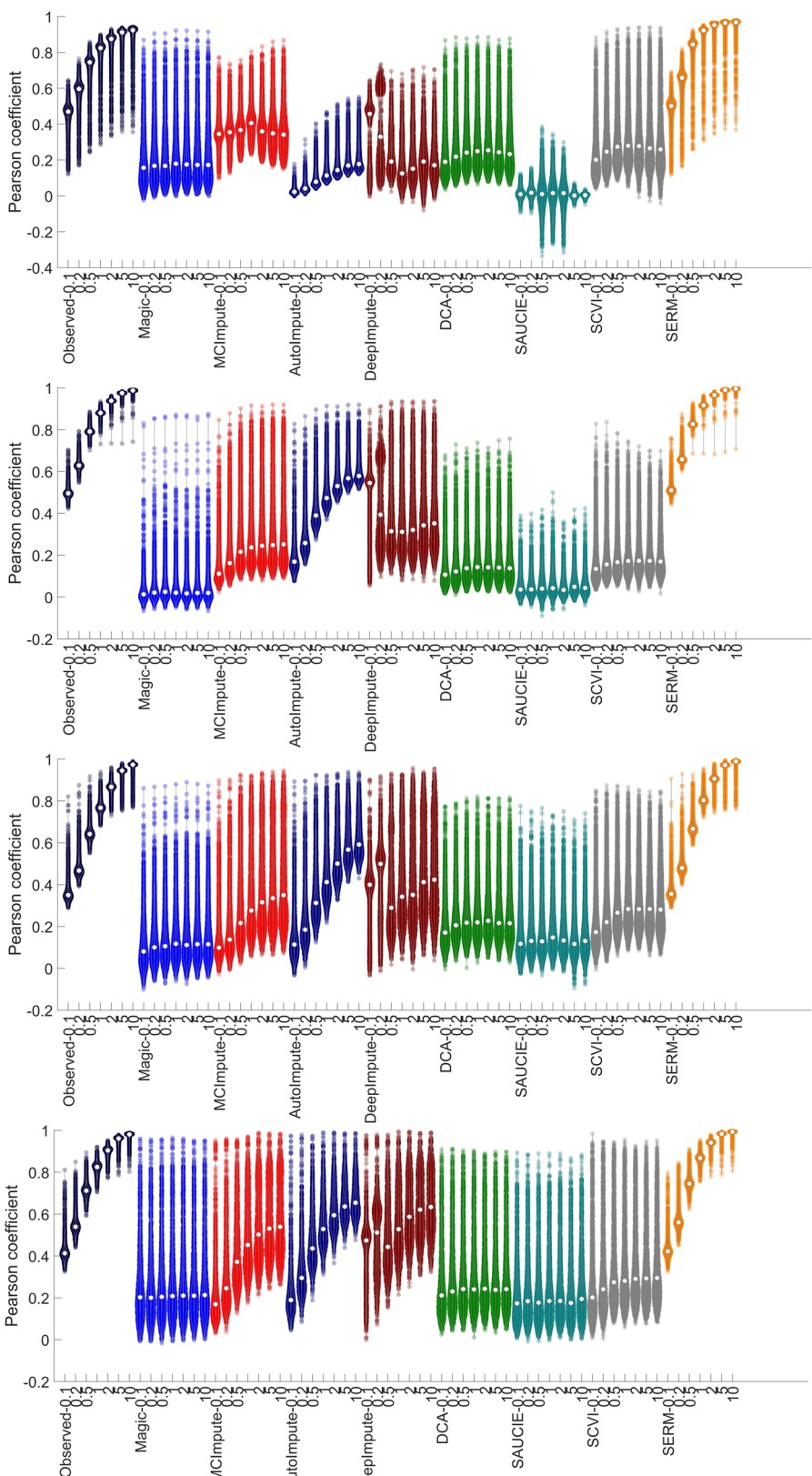

**Fig. 4 | Pearson coefficient between the reference and imputed data by eight different techniques for different datasets (cellular taxonomy-row 1, mammalian brain-row 2, mouse intestinal epithelium-row 3, and 3D neural tissue data-4th row).** The sampling efficiency (0.1–10%) to create the observed data are shown in x-axis. The center of the violin (denoted with a white circle) denotes the median value, and the spread of the violin denotes the standard deviation of the coefficient values across different cells. Source data are provided as a Source Data file.

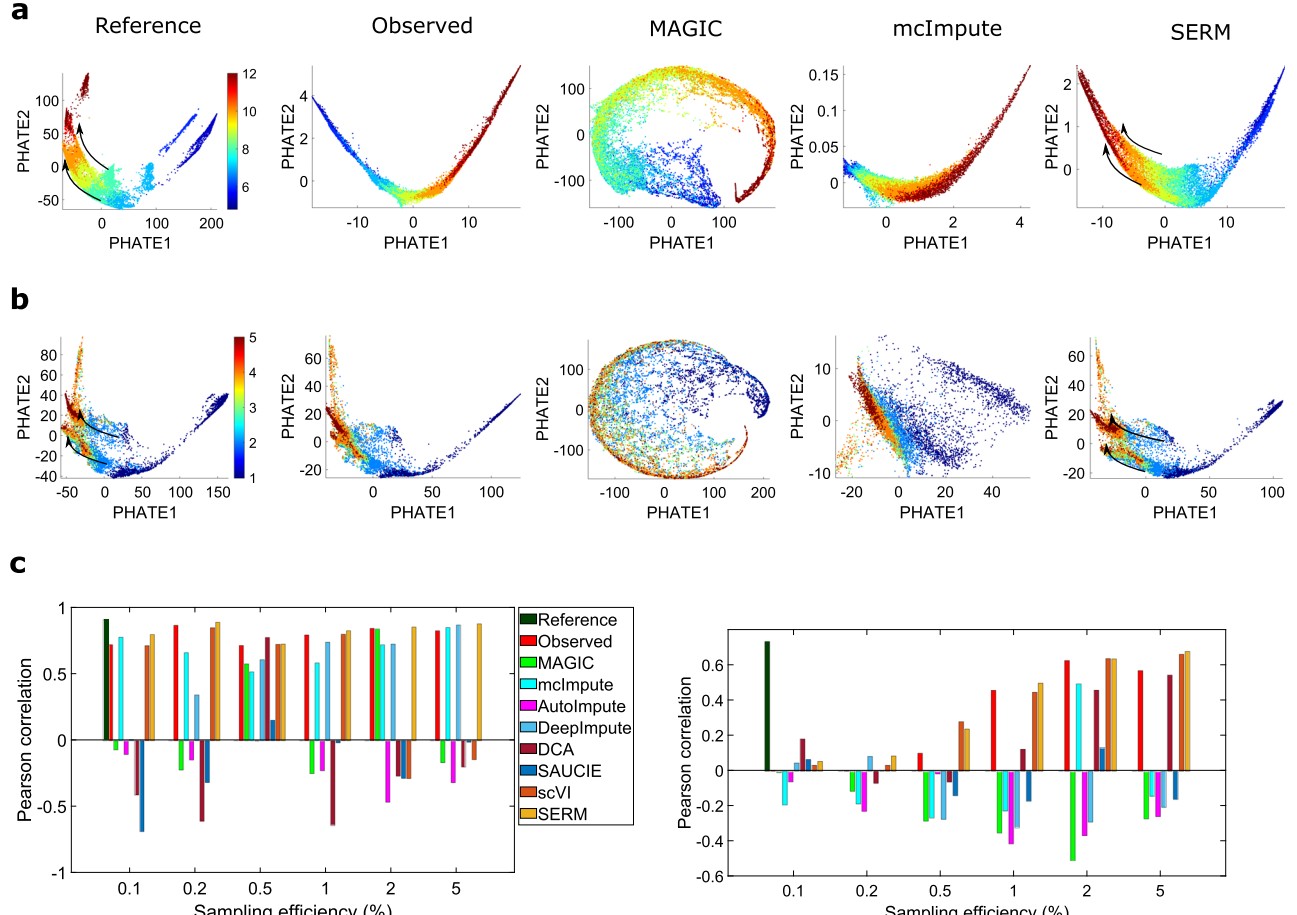

**Fig. 5 | Analysis of cell trajectory in imputed data from different methods.**
PHATE results from the reference data (first column), observed data (second column), imputed data from MAGIC, mcImpute, and SERM (columns 3–5) for **a** zebrafish development data and **b** EB differentiation data. The observed data were created by sampling the reference data at 0.1% efficiency for both datasets. All the trajectories are better visualized in SERM imputed data. MAGIC and mcImpute distort the data in both cases, whereas SERM retains the consistency of the data intact in both cases. The colorbar for **a** denotes the hpf (hours post fertilization). The colorbar of **b** represents 1-(0–3 days), 2- (6–9 days), 3- (12–15 days), 4- (18–21 days) and 5- (24–27 days)). Pearson coefficient between the pseudotime estimated by monocle from the imputed data and the data labels for all the methods are shown for zebrafish development data (left), and EB differentiation data (right) in **c**. Source data are provided as a Source Data file.

performance of MAGIC, DeepImpute, and mcImpute are comparable to SERM only in a few cases.

SERM is computationally much more efficient than other techniques. For example, SERM can analyze data with 0.6 million cells and 30,000 genes in <18 min on the PC mentioned earlier. In contrast, MAGIC, mcImpute, AutoImpute, and DeepImpute cannot complete the data analysis within 3 days. The computational speeds of different techniques for different data sizes are reported in the Supplementary Fig. S22, Tables 1 and 2. The unprecedented enhancement in computational efficiency arises from SERM using a fast histogram equalization operation to impute the data. Thus, SERM does not need computationally intensive calculations, such as determining the similarity between cells or genes. We note that SAUCIE is computationally more efficient than other methods except SERM.

## Discussion

Genomic sequencing techniques are now increasingly focused on the characterization of single cells. For many practical applications, such as data dimensionality reduction, visualization, and cellular spatial and temporal mapping, rectifying the expression values via reliable imputation is a prerequisite. In reality, however, the existing gene imputation methods suffer from several drawbacks: (1) a specific model for gene expression values is often assumed, which may not reflect the actual data distribution in many practical cases; (2) the data

is imputed based on the similarity among a few cells or genes, which is susceptible to errors as the global relationship among the cells and genes is not leveraged; (3) all zero expression values are often viewed as dropout events, which may not be accurate in practical settings; and (4) a priori information about the data is required to tune the hyperparameters of some methods. The proposed SERM mitigates these limitations through an effective learning strategy and significantly improves the imputation accuracy. It should be noted that SERM also assumes specific distributions for the expression data (such as exponential, Gaussian, and Rayleigh). However, the distribution parameters are learned in SERM, which ensures proper model adaptation to the data.

Data distortion, a condition when dropout positions in an expression matrix are filled with inaccurate values, is one of the main concerns in gene imputation. Preventing distortion is critically needed in applications like trajectory mapping, where the accurate position of each cell is required to find the trajectory continuum. In our study, we found that most methods, except SERM, distorted the data in many cases and could not reliably uncover the underlying biological information. With the learning of the data distribution, SERM shows remarkable performance in imputing the gene expressions for trajectory inference, as revealed in our extensive experiments. The technique consistently outperforms the existing methods for low and high dropouts.

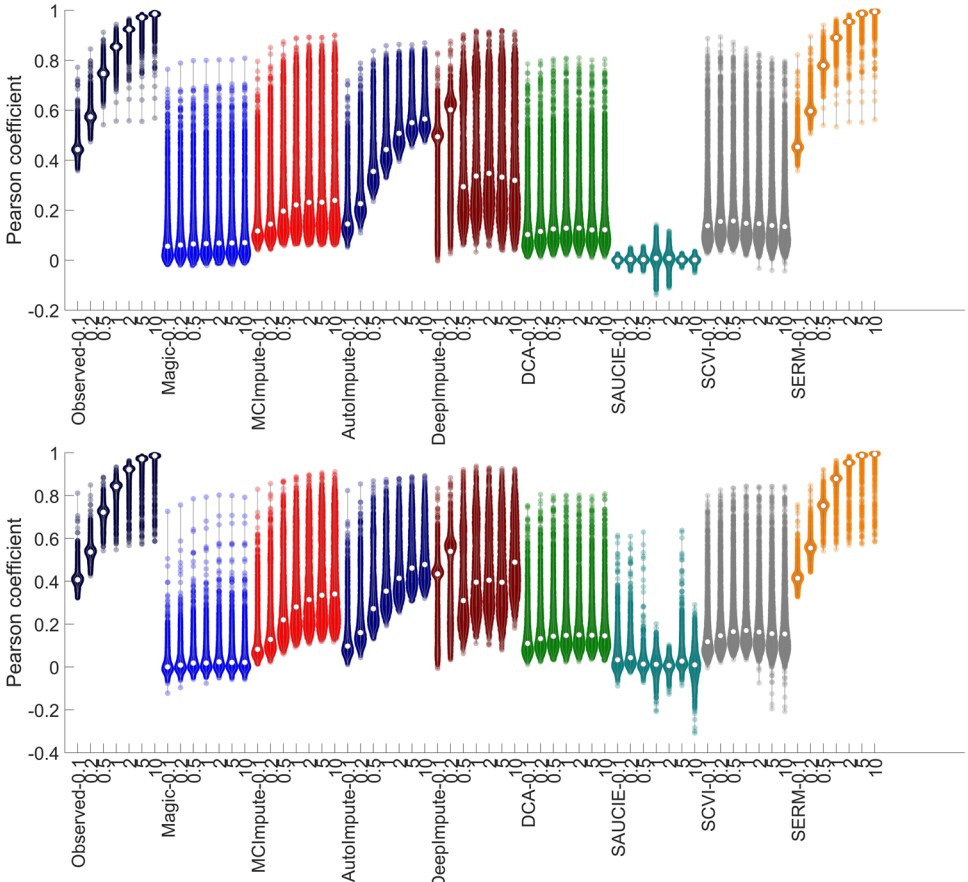

**Fig. 6 | Pearson coefficients between the imputed and reference data for all the methods are shown for zebrafish development data (first row), and EB differentiation data (second row).** The sampling efficiency (0.1–10%) to create the observed data are shown in *x*-axis. The center of the violin (denoted with white circle) denotes the median value and the spread of the violin denotes the standard deviation of the coefficient values across different cells. Source data are provided as a Source Data file.

In SERM, a data-driven histogram equalization is performed on each ROI to impute the gene expression values. Although histogram equalization is a popular technique in image processing for contrast enhancement, the assumption of a gene expression matrix as an image with ordered data elements is not necessary here. An implicit assumption in our formulation is that the coverage of ROI is sufficiently large so that the captured data distribution in the ROI is representative of the entire matrix. As only the distribution of expression values in a sufficiently large ROI is considered, the order of rows and columns is generally not important. In other words, if a gene (column) or a cell (row) is interchanged with another one, the distribution of the gene expression values and the resultant imputation should not change much. To ensure unbiased data distribution in different parts of the matrix, an optional step in which the columns/rows can be permuted randomly is included in our implementation. However, in most cases, this randomization step is unnecessary as the genomic data are generally not ordered. To illustrate this, we computed the correlation values from SERM imputed data for more than twenty expression datasets with 1000 different random alterations of cells and genes (Supplementary Fig. S29 shows an example for one of the datasets). We found that the correlation values remain almost identical (Supplementary Fig. S29), showing that SERM is robust against random changes in the positions of the genes and cells. We have also shown examples, where the dataset is arranged based on cell type or only one type of cell is considered (Supplementary Figs. S25 and S26). However, even in these exceptional cases, the accuracy of SERM drops by only <2% and the method can still be employed for high-performance gene imputation. The reason behind

this is that, while the distribution of gene expression changes from cell type to cell type, the change is not very significant (see Supplementary Fig. S27, where we show the gene expression distribution for ten different cell types of a dataset). It is seen from Supplementary Fig. S27 that, in all cell types, the distribution of gene expression has a similar curve with slight differences in variance and mean.

We have considered three analytical distributions (Gaussian, Rayleigh, and exponential) to model the data in SERM. Other distributions can also be included in SERM (see Python/Matlab codes of SERM). One can also use empirical distribution to characterize the data. In Supplementary Fig. S28, we have included SERM-imputed results for one dataset when the empirical distribution is used in place of the analytical distributions. It is seen that the SERM with an empirical distribution performs inferiorly to that with learned analytical distributions. For ultra-large data (e.g., on a scale of millions), the empirical distribution may characterize the data better and lead to improved performance. In SERM, we have used adaptive histogram equalization to perform the histogram correction of the dropout-affected data. However, other histogram matching techniques, such as quantile normalization[45] can also be used. We have added results of SERM with quantile normalization for the cellular taxonomy dataset, where it is seen that the results degrade. The performance degradation is because of the inferior performance of quantile normalization compared to adaptive histogram equalization[46].

Batch-effect correction plays a vital role in the scRNA-seq data analysis pipeline. A few methods that can perform both batch-effect correction and data imputation in the same calculation process, such as scVI[26] and CarDEC[47], have been reported. Although batch effects are

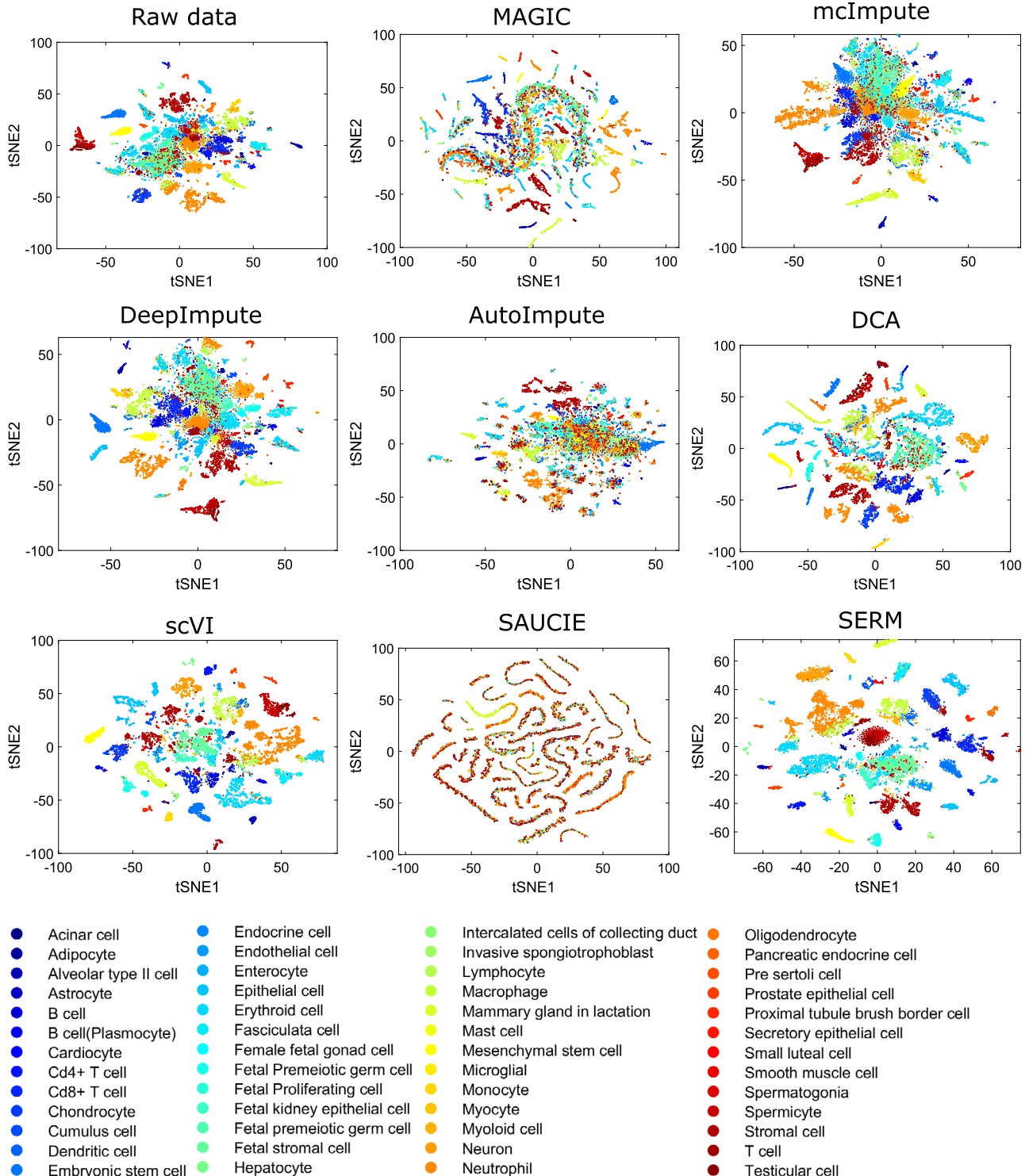

**Fig. 7 | t-SNE visualizations of the raw and imputed data by eight different methods for the MCA data.** Source data are provided as a Source Data file.

not considered explicitly in the formulation, SERM is generally less susceptible to data obtained using different technologies or protocols (see Supplementary Figs. S44–S46 for benchmarking of SERM on datasets with batch-effect). This is because SERM enforces a learned distribution from a single set of data (e.g., a batch of data) to all the data from different batches. Even if the data of different batches have different distributions, SERM forces them to follow the learned distribution, alleviating potential bias and artifacts (see "Methods" section).

As discussed in the introduction, a deep learning model generally proceeds in a black-box fashion with little control given to the user. In SERM, only the data distribution is learned via deep learning, and the actual imputation is performed by using an analytical histogram matching technique. Thus, the method is more transparent than traditional deep learning-based imputation techniques. We note that, in SERM, the distribution can also be preset by the user, which makes it easy for a user to investigate the effect of the learned distribution on the imputation. This also empowers the user to examine the results in

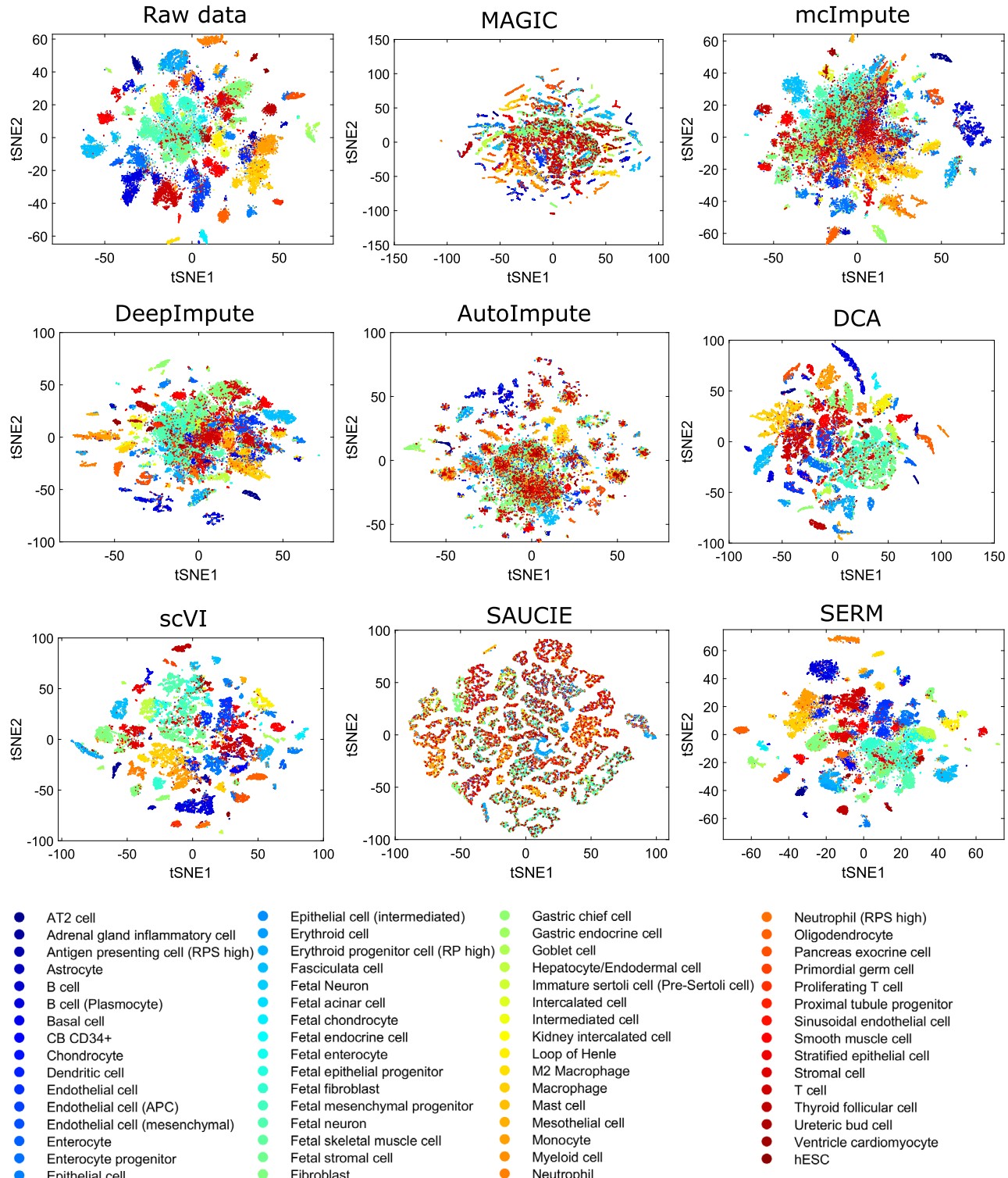

**Fig. 8 | t-SNE visualizations of the raw and imputed data by eight different methods for the HCL data.** Source data are provided as a Source Data file.

depth when a question arises about the modeling, which is not possible in other deep learning-based imputation techniques.

In DCA, SAUCIE or other encoder-based imputations, the objective function is based on reconstructing the original data from the latent representation. The approach affords a unique way to denoise the data. However, the methods do not put any constraint on the distribution of the data. By leveraging an analytical distribution to impute the values, SERM denoises the data and imposes natural guidance on the data distribution, leading to substantially improved

imputation accuracy and efficiency. While it may be possible to create an autoencoder with constraints on the data distribution, computationally, the approach would be much less efficient than SERM, especially when the data size increases.

SERM is scalable to data size and dimensionality, making it possible to impute large data with high computational efficiency. The scalability of SERM has been demonstrated by imputing two large datasets (HCL and MCA). In the visualizations of projected data in a low dimension, many new cell clusters are found after SERM imputation.

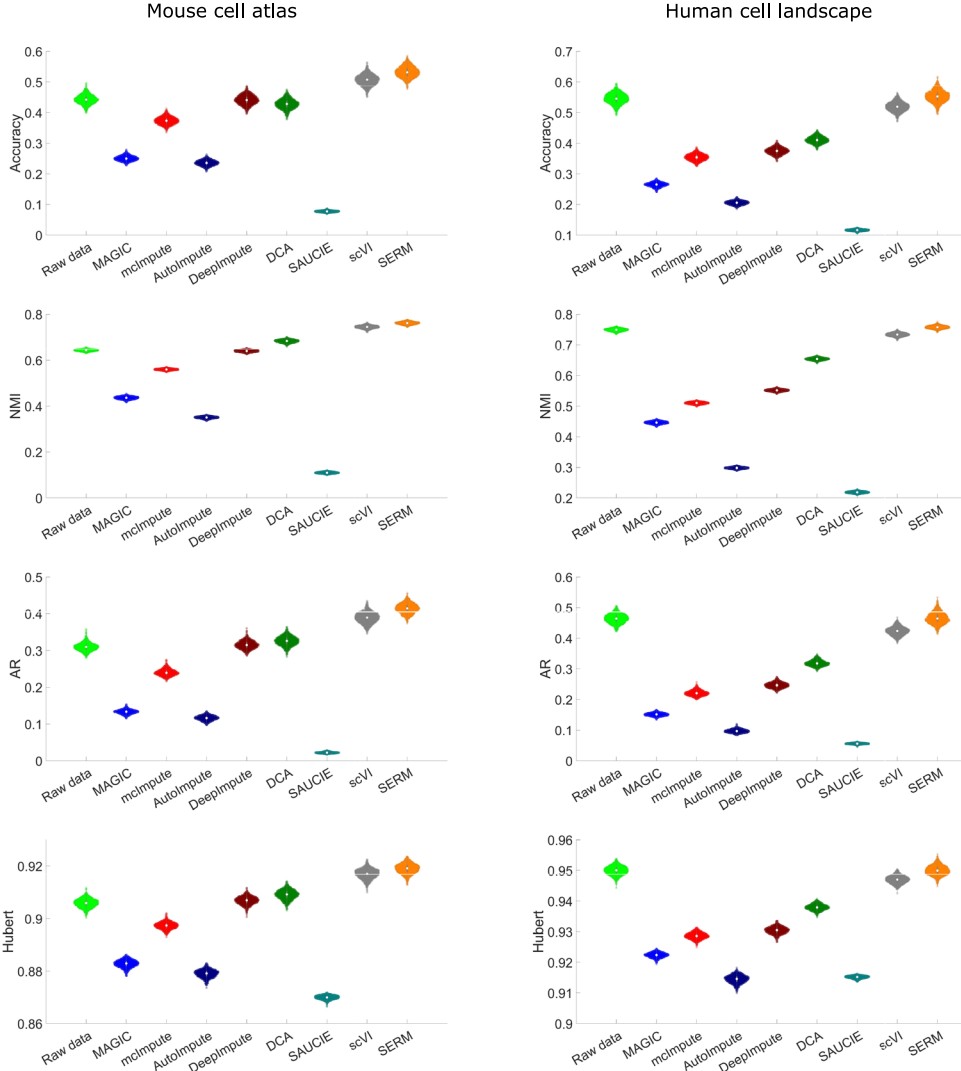

**Fig. 9 | Accuracy and three cluster quality indices (NMI, AR, and Hubert) computed from t-SNE results of the raw and imputed data by different methods for the MCA and HCL data.** Source data are provided as a Source Data file.

Thus, using SERM, it may be possible to discover novel biological information, such as cell or gene types, from large scRNA-seq data.

In the evaluation of SERM, the reference data created by choosing only the cells and genes with high expressions provides a dropout-free representation. The observed data is created from the reference data by simulating efficiency loss that introduces zeros and represents the dropout-affected data. Different dropout rates were simulated by varying the parameters of Gamma distribution. The applications of SERM on a large number of datasets with various dropout rates show its broad applicability. Different methods are followed in single cell research community to create the dropout-affected data[12,17,18]. To prove the versatility of SERM in single-cell imputation, we have added Supplementary Figs. S51 and S52 showing the performance of SERM for three different techniques of creating the dropout-affected data[12,17,18]. It is seen that, in all cases, SERM can improve the data quality (which no other method can).

We evaluated SERM on seventeen different scRNA-seq datasets from different biological systems and measurement technologies. In all cases, SERM accurately recovers data clusters (Figs. 2–4, 7–9, Supplementary Figs. S49, S50), cell developmental trajectories, and state transitions (Figs. 5, 6). The superior performance of SERM across all these datasets demonstrates its versatility in genetic engineering and computational biology. Generally, the issue of gene expression imputation here is a special case of missing data or missing value problems,

which can occur in almost all measurement modalities. In practice, the strategy of dealing with the problem can significantly affect the conclusions to be drawn from the data[48–50]. Since SERM is based on learning from the data, it is not restricted to a specific type of data. The method can thus adapt to new data easily and provides a broadly applicable strategy for missing data recovery across different disciplines. For this reason, SERM is equally applicable for both raw and normalized (using library size, logarithm, or other nonlinear operations) scRNA-seq datasets (see Supplementary Fig. S24 for an example). In addition to the missing value imputation, we emphasize that the proposed SERM also provides an effective solution in denoising the data, as demonstrated in Figs. 2–9.

In our analyses, we observed that MAGIC distorts the t-SNE/UMAP visualizations of the data in many cases. We emphasize that this may not imply the bad quality of the MAGIC imputation, as the distortions may also arise from the visualization tools. Rather, we found that MAGIC performs better than many techniques in terms of quantitative indices such as clustering accuracy, NMI, and cluster quality indices. This phenomenon is corroborated by an earlier study by Hou et al.[15].

Despite its success, SERM is not without limitations. In particular, SERM learns the gene expression model from a small fraction of the data. This poses a challenge in dealing with small genomic datasets, as the estimated distribution in SERM may diverge from the actual data distribution, leading to inaccurate imputation values.

In summary, we have proposed a novel computational strategy for gene expression recovery. The technique can impute gene expression data of different levels of dropout rate with unprecedented accuracy, reliability, and consistency. Going beyond traditional imputation techniques, SERM learns the data distribution and applies the knowledge to the subsequent calculation under the condition of self-consistency. Apart from the computational efficiency and scalability, SERM substantially improves the performance of the imputation process with augmented interpretability in comparison to the existing deep learning techniques. As SERM is data-driven and can adjust to any data type, we envision that SERM will play an important role in various biomedical and biocomputational applications.

## Methods

The SERM strategy involves the following major steps:

1. Modeling (Fig. 1 steps 1–3): The goal of this step is to determine the distribution that best represents the gene expression data. First, a subset of the original dataset is defined. This data subset is transformed into a denoised version of itself via an unsupervised deep learning technique that performs compression and decompression of the data. This process removes noise and retains the essential latent features of the data, thus outputting a denoised representation of the original data. A histogram of the denoised expression values is computed, and distribution fitting is performed on the histogram to output the learned pdf.
2. Imputation (Fig. 1 steps 4–6): The values in the gene expression matrix are systematically imputed based on a sliding window. A rectangular ROI is defined with a width and height smaller than the width and height of the expression matrix. The ROI serves as a window into the matrix—only the values within the window are manipulated in a single instance. The sliding parameters, including the ROI and step sizes, are then defined to determine the distance that the window shifts for each iteration. For each window position, a histogram of the values in the window is computed. This histogram is equalized to the histogram of the denoised expression values found in the previous step. The new window values found through histogram equalization substitute for the original values. The window continues sliding until the entire matrix is imputed.
3. Consistency (Fig. 1 step 7): If the sliding distance of the window is smaller than the window size, then the values in each window overlap with those in the adjacent windows. Cross-window consistency is automatically imposed for this choice of parameters. If the sliding distance is equal to the window size, adjacent windows do not overlap, and the values in each window are imputed independently from the rest of the matrix. Bilinear resampling is performed to impose global consistency across all the independent windows for this case.

The details of each step of SERM are discussed below.

### Learning data distribution

Deep neural networks (DNNs) have shown promising results in classifying biomedical data[51]. Autoencoders[52] (consists of an encoder and a decoder) are a class of DNNs for data compression in which low dimensional latent features are learned from high dimensional data. It learns only the essential latent features while ignoring non-essential sources of variation such as random noise[22]. Therefore, the compressed latent data of the network is a representation of the high dimensional ambient data space in lower dimensionality and captures the underlying true data manifold. Thus, when the autoencoder reconstructs the high-dimensional data, it represents the ideal version of the high-dimensional denoised data[22]. We use a sub-matrix ($S_s$ of size $q_s \times m_s$) of the expression data ($S$ of size $q \times m$) as the input to the autoencoder network to learn the distribution of the denoised expression data.

### Denoising via auto-encoding.

The deep learning architecture (auto-encoder) used in SERM consists of two parts, an encoder and a decoder. The encoder of the network takes the input $\boldsymbol{x}_d \in \mathbb{R}^{m_s}$ ($\boldsymbol{x}_d$ is a row of $S_s$) and maps it to a latent space $\mathbf{h} \in \mathbb{R}^P$, where $m_s > P$ and $\mathbf{h}$ is defined as

$$\mathbf{h} = \sigma(\mathbf{W}\mathbf{x_d} + \mathbf{b}). \tag{1}$$

Here, $\sigma$ is an element-wise activation function which can be a linear or nonlinear (such as sigmoid) function. $\mathbf{W}$ and $\mathbf{b}$ are the weight matrix and bias vector, respectively. The decoder of the network maps the latent variable $\mathbf{h}$ to the reconstructed data $\mathbf{x'_d}$. The equation for $\mathbf{x'_d}$ can be written as

$$\mathbf{x'_d} = \sigma'(\mathbf{W'}\mathbf{h} + \mathbf{b'}). \tag{2}$$

Here $\sigma'$, $\mathbf{W'}$ and $\mathbf{b'}$ are the activation function, weight matrix, and bias vector for the decoder. In the training phase, the weights and biases are initialized randomly and updated iteratively using the back-propagation technique. During training the network, the reconstruction error (mean squared error) is minimized. In particular, the mean squared error (also called loss) can be expressed as

$$\mathcal{L}(\mathbf{x_d}, \mathbf{x'_d}) = \| \mathbf{x_d} - \mathbf{x'_d} \|^2. \tag{3}$$

$\mathbf{x'_d}$ are the rows of reconstructed output matrix $S_o$, which is of the same size as $S_s$.

### Histogram computation.

Consider the gene expression matrix $S_o = [s_{ij}] \in \mathbb{R}^{q_s \times m_s}$, where $q_s$ is the number of cells and $m_s$ is the number of genes. For each expression value, the number of elements with that value in $S_o$ is counted. The collection of these counts for all the expression values is referred to as the histogram of that region. This function is an empirical estimate of the expression density function. Let $N_{S_o}$ and $L_{S_o}$ denotes the number of matrix elements and possible expression values in $S_o$. Furthermore, we define the normalized histogram associated with the possible gene expression values in $S_o$ as follows

$$p_{S_o}(n) = \frac{N_{S_o}(n)}{N_{S_o}}, \quad n = 0,1,2,\cdots,L_{S_o} - 1, \tag{4}$$

where $N_{S_o}(n)$ is the number of elements in the ROI with the gene count $n$ such that $N_{S_o} = \sum_{n=0}^{L_{S_o}-1} N_{S_o}(n)$.

### Curve fitting.

After applying the histogram equalization, we fit different probability distribution functions (PDF) to $p_{S_o}$. Specifically, we consider

$$f_g(x) = \frac{1}{\lambda\sqrt{\pi}} e^{-(\frac{x}{\lambda})^2}, \tag{5}$$

$$f_r(x) = \frac{2x}{\lambda^2} e^{-(\frac{x}{\lambda})^2}, \tag{6}$$

$$f_e(x) = \lambda e^{-\lambda x}, \tag{7}$$

where $f_g, f_r$ and $f_e$ denote Gaussian, Rayleigh and exponential PDFs. The PDF with parameter $\lambda$ that results in minimum root mean square (RMSE) is chosen for next steps of SERM. We note that although we used three PDFs in our analysis, other PDFs (such as log-normal, student-t and gamma) can be easily incorporated into SERM implementation (see implementation codes). Below we describe the mathematical analysis for histogram equalization of exponential PDF. Analysis for other PDFs can be performed in a similar manner.

## ROI selection and histogram equalization

We divide the expression matrix $S$ is divided into $N \geq 4$ ROIs, which we denote by $S_i$, $i = 1, 2, \cdots, N$. Let $N_{S_i}$ and $L_{S_i}$ denotes the number of matrix elements and possible expression values, in each ROI $S_i$, respectively. Furthermore, we define the normalized histogram associated with the possible gene expression values in the ROI $S_i$ as follows

$$p_{S_i}(n) = \frac{N_{S_i}(n)}{N_{S_i}}, \quad n = 0, 1, 2, \cdots, L_{S_i} - 1, \tag{8}$$

where $N_{S_i}(n)$ is the number of elements in the ROI with the gene count $n$ such that $N_{S_i} = \sum_{n=0}^{L_{S_i}-1} N_{S_i}(n)$. The corresponding cumulative distribution function (CDF), is given by

$$F_{S_i}(n) = \frac{1}{N_{S_i}} \sum_{k=0}^{n} p_{S_i}(k), \quad n = 0, 1, 2, \ldots, L_{S_i} - 1. \tag{9}$$

**Adaptive equalization of histogram.** The goal of the histogram adaptive equalization is to find a transformation for the expression values such that the transformed CDF is approximated by the CDF of $f(x)$, where $f(x)$ can be $f_g(x)$, $f_r(x)$ or $f_e(x)$. As example, the CDF of an exponential distribution can be written as

$$F(t) = \begin{cases} 1 - e^{-\lambda t}, & t \geq 0 \\ 0, & t < 0 \end{cases} \tag{10}$$

where $\lambda$ denotes the decay parameter. SERM uses the transformation $T_{S_i} : \{0, 1, \cdots, L_{S_i} - 1\} \rightarrow \{0, 1, \cdots, L_{S_i} - 1\}$ for each ROI to map the expression values to their new values. In particular, consider transforming the expression counts by

$$T_{S_i}(k) = -\left\lfloor \frac{1}{\lambda} \ln\left(1 - F_{S_i}(k)\right) \right\rfloor, \tag{11}$$

where the floor function $\lfloor x \rfloor$ is the biggest integer not exceeding $x$. The map $T_{S_i}(k)$ is then applied to each expression value in the ROI $S_i$.

The main intuition behind this transformation comes from the inverse sampling method. In particular, let $X$ denotes continuous random variables whose CDF is strictly increasing on the possible value on $[0, L_{S_i} - 1]$ with densities $p_X$ and $p_Y$, respectively. Consider the transformation

$$Y = T_0(X) = (L_{S_i} - 1) \int_0^X p_X(u) \, du. \tag{12}$$

Then, $Y$ has the uniform distribution, i.e., $Y \sim \text{Uniform}[0, L_{S_i} - 1]$. To see this note that

$$\begin{aligned} P\{Y \leq y\} &= P\{(L_{S_i} - 1) F_X(X) \leq y\} \\ &= P\left\{X \leq F_X^{-1}\left(\frac{y}{L_{S_i} - 1}\right)\right\} \\ &= F_X\left(F_X^{-1}\left(\frac{y}{L_{S_i} - 1}\right)\right) \\ &= \frac{y}{L_{S_i} - 1}. \end{aligned} \tag{13}$$

Furthermore, using the inverse transform sampling technique, the random variable

$$Z = T_1(X) = -\frac{1}{\lambda} \ln(1 - Y/(L_{S_i} - 1)), \quad Y \sim \text{Uniform}[0, L_{S_i} - 1], \tag{14}$$

has the exponential distribution, i.e., $Z \sim F(t)$. Combining the transformations $T_0$ and $T_1$, we conclude that

$$Z = T_1(T_0(X)) = -\frac{1}{\lambda} \ln\left(1 - \int_0^X p_X(u) \, du\right), \tag{15}$$

has the desired exponential distribution. Now, the transformation $T_{S_i}$ defined in Eq. (11) is an inverse sampling method for the discrete random variable similar to the continuous transformation $T_1 \circ T_0$. Similar analytical equations to Eqs. (10)–(15) can be derived for Gaussian and Rayleigh distributions.

## Sliding ROI

The ROI is slided throughout the whole expression matrix and gene imputation is performed at each ROI to obtain the resulting imputed gene expression matrix from SERM.

## Resampling for patch consistency

In the previous step, the imputation is performed on each ROI by matching the transformed clipped histogram of each time with a distribution. We now incorporate the histograms of different ROIs into a consistent framework by applying a bilinear resampling. Let $\widehat{S} = [\widehat{s}_{ij}] \in \mathbb{R}^{q \times m}$ denotes the gene expression matrix after the imputation of each ROI via the histogram clipping and equalization step. We subsequently resample each expression value $\widehat{s}_{ij}$ using the values of lateral elements $Q_{11} = \widehat{s}_{(i-1)(j-1)}$, $Q_{12} = \widehat{s}_{(i-1)(j+1)}$, $Q_{22} = \widehat{s}_{(i+1)(j+1)}$, and $Q_{21} = \widehat{s}_{(i+1)(j-1)}$. In particular, let $\bar{s}_{ij}$ denotes the resampled value which is computed by the following sample average

$$\bar{s}_{ij} = \frac{1}{4}\left(\widehat{s}_{(i-1)(j-1)} + \widehat{s}_{(i-1)(j+1)} + \widehat{s}_{(i+1)(j+1)} + \widehat{s}_{(i+1)(j-1)}\right). \tag{16}$$

For the boundary elements of the matrix $\widehat{s}_{i1}$, $\widehat{s}_{iq}$, $\widehat{s}_{1j}$, and $\widehat{s}_{nj}$ the above formula is used in conjunction with the zero padding of the matrix $\widehat{s}_{i0} = \widehat{s}_{0j} = \widehat{s}_{i(q+1)} = \widehat{s}_{(n+1)j} = 0$ for all $i \in \{1, 2, \cdots, q\}$ and $j \in \{1, 2, \cdots, m\}$. The interpolation formula in Eq. (16) can be dervied via the discretization of the bilinear interpolation of continuous functions of two variables. The resulting matrix $\bar{S} = [\bar{s}_{ij}] \in \mathbb{R}^{q \times m}$ from resampling in Eq. (16) yields the desired imputed matrix.

## Simulation procedures

All the simulated data were created in R language using Splatter simulator. The method of simulation was set to 'groups' and the probability of each of the five groups was set to 0.2. All other simulation parameters were set to default values.

## Implementation and parameter settings

Python and Matlab (MathWorks Inc., Natick, MA, USA) implementations of the SERM technique were performed. Although the whole dataset can be used for learning the distribution of denoised data, to reduce the computational burden, 2000 data points were randomly selected (see supplementary Table 3, Figs. S30–S32 for detailed analysis). For all the analyses, a three-layer (input, latent, output) network was used in the deep learning step of SERM. A linear activation function was used as the decoder transfer function and a logistic sigmoid function as the encoder transfer function[52]. The value of $P$ (dimension of latent space) in the encoder-decoder network was set to 2. L2 and sparse regularizers were used with parameter values of 0.05 and 0.9. The learning rate was fixed at $1e-5$, and the max epoch was set to 20 with a batch size of 64. The adam optimizer was used to train the model. The neuron values were initialized randomly from a uniform distribution in the range of 0–1.

The 'Trust-Region' algorithm was used for curve fitting with mean square error as the cost function. For Gaussian and Rayleigh equation

fitting, the starting point, lower limit, and upper limits of $\lambda$ were 0.2, 0.01, and 1, respectively. For exponential equation fitting, the starting point, lower limit, and upper limits of $\lambda$ were 5, 5, and 20, respectively. These values were chosen based on fitting the equations to ideal data generated using the Splatter simulator. In adaptive histogram equalization of SERM, the window length was set to half of the matrix length, and the window height was set to half of the matrix height (see supplementary Fig. S23 for results of SERM with quantile normalization). The x- and y-sliding parameters were set to the length and height of the window. We add analyses of different ROI sizes and overlaps in Supplementary Figs. S33 and S34, where it is seen that SERM is very robust to change of these parameters and provide accurate results for a large range of their values. For HCL and MCA datasets, the whole datasets were divided into 48 (24 by 2) and 28 (14 by 2), respectively, for SERM-imputation. For creating the histograms, we set the number of bins equal to 100.

In t-SNE and UMAP visualizations of data from all the methods in Fig. 2, the first 50 principal components (PCs) were used. In visualizations of Fig. 3, the first 20 PCs were used. For zebrafish data, 100 PCs and EB data, 50 PCs were used in PHATE for creating the visualizations in Fig. 5. The first 500 PCs were used to create the t-SNE visualizations of HCL and MCA datasets (Figs. 7 and 8).

UMAP implementation from the original authors was used with default parameters[53]. PHATE was used with default parameters: number of (output) embedding dimensions equal to 2, number of nearest neighbors equal to 5, an alpha value equal to 40, automatically determined the optimal number of diffusion steps ($t$), maximum $t$ equal to 40, number of principal components equal to 100, 'metric MDS' as the method of multidimensional scaling, euclidean as the distance function, log as the method of computing the PHATE potential distance, gamma value equal to 0.5, epsilon value equal to $1e-7$, number of landmarks equal to 2000, and number of singular vectors for spectral clustering equal to 100. Monocle, Slingshot, and TSCAN methods were applied to the datasets using default settings provided by the authors. The computational speed of SERM implemented in Matlab was used to benchmark its computational efficiency against other methods.

### SERM on data from different batches

If a dataset consists of data from different batches, data from any of the batches can be used as the reference. It should be noted that setting one batch-data as reference is a common practice in batch-effect correction techniques[54–56]. Indeed, for the examples presented in Supplementary section 11, we found that the results of SERM imputation change little when a different batch is used as the reference. The data distribution in SERM is learned from the reference batch and then applied to all batches for data imputation. In this case, the randomization of cells and genes, as mentioned in the Discussion section, is performed within their own batches before imputation calculation in SERM. After imputation, all the cells and genes are relocated to their original places. The SERM-imputed datasets from different batches are then integrated into a single dataset using a z-score operation for downstream analyses. Please see supplementary Fig. S43 for a detailed workflow of SERM when a dataset contains data from different batches.

### Datasets

The following datasets were used to benchmark SERM against different existing imputation techniques:

**Cellular taxonomy of the mouse bone marrow stroma.** In this dataset, scRNA-seq was used to define a cellular taxonomy of the mouse bone marrow stroma, and its perturbation by malignancy[36]. Seventeen stromal subsets were identified expressing distinct hematopoietic regulatory genes spanning new fibroblastic and osteoblastic subpopulations including distinct osteoblast differentiation trajectories. Emerging acute myeloid leukemia impaired mesenchymal osteogenic differentiation and reduced regulatory molecules necessary for normal hematopoiesis. This taxonomy of the stromal compartment provides a comprehensive bone marrow cell census and experimental support for cancer cell crosstalk with specific stromal elements to impair normal tissue function and thereby enable emergent cancer.

**Mammalian brain.** The dataset is from a massively parallel scRNA-seq technology namely sNucDrop-seq (single-nucleus RNA-seq approach)[37], which provides unbiased isolation of intact single cells from complex tissues such as adult mammalian brains. The authors profiled 18,194 nuclei isolated from cortical tissues of adult mice. The authors demonstrated through extensive validation that sNucDrop-seq not only accurately reveals neuronal and non-neuronal subtype composition with high accuracy but also allows in-depth analysis of transient transcriptional states driven by neuronal activity.

**Mouse intestinal epithelium.** Intestinal epithelial cells absorb nutrients, respond to microbes, function as a barrier, and help to coordinate immune responses. 53,193 individual epithelial cells from the small intestine and organoids of mice were profiled by Haber et al.[38], which enabled the identification and characterization of previously unknown subtypes of intestinal epithelial cell and their gene signatures.

**Human-engineered neural cells.** The dataset is from a study on human-engineered neural tissues[39]. ScRNA-seq data of human-induced neuronal cells were cultured in two different conditions: (1) with mouse astrocytes or (2) with differentiated human astrocytic cells. We analyze the data from the second condition, where hESC (human embryonic stem cells) induced neuronal cells and human astrocytic cells differentiated from hESCs were co-cultured at a 1:1 ratio in a 3D composite hydrogel.

**Zebrafish embryogenesis.** The dataset[41] was profiled using a massively parallel scRNA-seq technology named Drop-seq[2]. Data were acquired from the high blastula stage (3.3 hours postfertilization (hpf), moment after transcription starts from the zygotic genome) to six-somite stage (12 h after postfertilization, just after gastrulation). Most cells are pluripotent at the high blastula stage, whereas many cells have differentiated into specific cell types at the six-somite stage.

**EB differentiation.** EB differentiation recapitulates key aspects of early embryogenesis. It has been successfully used as the first step in differentiation protocols for certain types of neurons, astrocytes and oligodendrocytes, hematopoietic, endothelial and muscle cells, hepatocytes and pancreatic cells, and germ cells. Approximately 31,000 cells were measured, equally distributed over a 27-day differentiation time course by the authors of ref. 40. Samples were collected at 3-d intervals and pooled for measurement on the 10x Chromium platform[40,42].

**Human cell landscape.** The HCL is a basic landscape of major human cell types created based on samples from a Han Chinese population using Microwell-seq technology[57]. Donated tissues were perfused or washed and prepared as single-cell suspensions using specific standard protocols. The analyses included fetal and adult tissue samples and covered 60 human tissue types. Seven types of cell culture, including induced pluripotent stem (iPS) cells, embryoid body cells, hematopoietic cells derived from co-cultures of human H9 and mouse OP9 cells[58], and pancreatic beta cells derived from H9 cells using a seven-stage protocol were also analyzed[59]. Single cells were processed using Microwell-seq[60] and sequenced at around 3000 reads per cell;

data were then processed using published pipelines[8]. In total, 702,968 single cells passed the quality control tests (please see ref. 57 for details of the quality control tests). Following the authors' work, we used 599,926 cells with 63 unique cell types and 59 unique tissue types for our analysis.

**Mouse cell atlas.** For creating the MCA[60], Mammary gland (virgin, pregnant, lactation and involution), uterus, bladder, ovary, intestine, kidney, lung, testis, pancreas, liver, spleen, muscle, stomach, bone marrow, thymus, prostate, cKit+ bone marrow, bone marrow mesenchymal cells and peripheral blood samples from 6- to 10-week-old C57BL/6 mice were collected. Single cells were then sequenced with Microwell- seq. The sequencing data were processed using published pipelines[8]. Following the work of the authors[60], we analyzed 333,778 cells with 52 unique cell types and 47 unique tissue types.

### Generating reference and observed datasets
To generate a reference dataset from real scRNA-seq data, following ref. 12, we selected high-quality cells and genes with high expressions from the original dataset to be the true expression $\lambda_{gc}$. We generated the observed datasets by drawing from a Poisson distribution with mean parameter $\tau_c\lambda_{gc}$, where $\tau_c$ is the cell-specific efficiency loss. We aimed to select roughly 10–20% of genes with the highest proportion of cells with nonzero expression and 50–60% of the cells with the largest library size. All the datasets were converted to transcripts per million (TPM) before filtering so that downsampled datasets for a range of efficiency losses (0.1–10%) can be created. The specific filters used for each dataset are described in supplementary section 12 (Supplementary Table 4, Figs. S47 and S48).

To mimic variation in efficiency across cells, we sampled $\tau_c$ as follows[12]:

1. 10% efficiency: $\tau_c \sim Gamma(10, 100)$
2. 5% efficiency: $\tau_c \sim Gamma(10, 200)$
3. 2% efficiency: $\tau_c \sim Gamma(10, 500)$
4. 1% efficiency: $\tau_c \sim Gamma(10, 1000)$
5. 0.5% efficiency: $\tau_c \sim Gamma(10, 2000)$
6. 0.2% efficiency: $\tau_c \sim Gamma(10, 5000)$
7. 0.1% efficiency: $\tau_c \sim Gamma(10, 10000)$

Detailed statistics of the reference and observed datasets at different sampling efficiencies are added in section 12 (Supplementary Figs. S47 and S48) of the supplementary. It is seen from Supplementary Fig. S47 that as the sampling efficiency reduces (from 10% to 0.1%), the number of zeros (denoting dropouts) increases, and vice versa. In Supplementary Fig. S48, it is seen that for all sampling efficiencies, the dropout probability increases for the smaller expression values.

### Competing methods
MAGIC was downloaded from https://github.com/pkathail/magic. mcImpute, DeepImpute, and SAUCIE were downloaded from https://github.com/aanchalMongia/McImpute_scRNAseq, https://github.com/lanagarmire/deepimpute, https://github.com/KrishnaswamyLab/SAUCIE. All these methods were used with default configurations.

### Computation of Pearson coefficient
Let us assume that $R_f$ and $R_e$ are the gene expression from dropout-free data and imputed data using different techniques. Let us assume, A is the dropout-free gene expression data for a single cell from $R_f$, whereas B is the gene expression vector for the same cell from $R_e$. Then the Pearson correlation coefficient between A and B is defined as

$$\rho(A,B) = \frac{1}{m-1}\sum_{i=1}^{m}\left(\frac{A_i - \mu_A}{\sigma_A}\right)\left(\frac{B_i - \mu_B}{\sigma_B}\right), \quad (17)$$

where $\mu_A$ and $\sigma_A$ are the mean and standard deviation of A, respectively, and $\mu_B$ and $\sigma_B$ are the mean and standard deviation of B.

Percentage improvement over observed data was defined as[12]

$$\% \text{ change over observed} = 100 \times \frac{\rho_{\text{method}} - \rho_{\text{observed}}}{\rho_{\text{observed}}}, \quad (18)$$

where $\rho_{\text{observed}}$ and $\rho_{\text{method}}$ are the mean Pearson coefficient of the observed data and imputed data by a method, respectively.

### Computation of NMI, accuracy, and cluster quality indices
We at first cluster the data into $N_g$ classes ($N_g$ is number of classes in ground truth label) by k-means clustering technique with Euclidean distance. Clusters were initialized using k-means++[61]. We then find the best map of cluster labels compared to the ground truth labels. These cluster labels are then used to compute the NMI, accuracy, and cluster quality indices: adjusted Rand (AR) and Hubert. NMI is the normalized mutual information[62] between the estimated labels and true labels computed following the work of Becht et al.[32]. Accuracy is the number of correctly found class labels divided by total number of class labels. Hubert and AR indices are computed using the formula reported in the work of Hubert et al.[34].

### Statistics and reproducibility
No statistical method was used to predetermine sample size. No data were excluded from the analyses. The experiments were not randomized. There was no blinding. The analyses performed do not involve evaluation of any subjective matters.

### Reporting summary
Further information on research design is available in the Nature Portfolio Reporting Summary linked to this article.

## Data availability
The datasets generated during and/or analyzed during the current study are available within the manuscript and supplementary. Source data are provided with this paper. Cellular taxonomy, mammalian brain, mouse intestinal epithelium, human-engineered neural cells, and zebrafish embryogenesis datasets were downloaded from Broad Institute single-cell portal (https://singlecell.broadinstitute.org/single_cell). EB differentiation data was obtained from the Github link: https://github.com/KrishnaswamyLab/PHATE. The human cell landscape and mouse cell atlas data were acquired from http://bis.zju.edu.cn/HCL/and http://bis.zju.edu.cn/MCA/. All other relevant data supporting the key findings of this study are available within the article and its Supplementary Information files or from the corresponding author upon reasonable request. Source data are provided with this paper.

## Code availability
SERM is available as a Code Ocean capsule (https://doi.org/10.24433/CO.7874136.v1). Its Python and Matlab source codes can be found at https://github.com/xinglab-ai/self-consistent-expression-recovery-machine(https://zenodo.org/badge/latestdoi/432331093)[63].

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

## Acknowledgements

This work was partially supported by NIH (1R01 CA223667-L.X. and R01CA227713-L.X.) and a Faculty Research Award (L.X.) from Google Inc.

## Author contributions

L.X. conceived the experiment(s), M.T.I. conducted the experiment(s), M.T.I., X.L., M.B.K., J.Y.W., L.Y., S.S., L.S., H.R., and W.Z. analyzed the results. All authors reviewed the manuscript.

## Competing interests

The authors declare no competing interests.
