## [Peer Review File · Nature Communications]

Leveraging data-driven self-consistency for high-fidelity gene expression recoveryReviewer #1 (Remarks to the Author):

Islam et al. developed SERM, a deep learning method that aims to impute gene expression in single-cell RNA-seq. Numerous methods have been developed to denoise or impute gene expression in scRNA-seq. The advantage of SERM compared to other methods is its speed. The authors showed that it has at least 100-fold increase in computational efficiency as compared to other methods. I have several major concerns about the paper.

1. The key idea of the paper is to consider gene expression matrix as an "image" and focus on gene expression imputation within each Region of Interest (ROI) and then slide the ROI along x and y direction throughout the entire gene expression matrix. Within each ROI, gene expression imputation is performed by histogram equalization and all ROIs are then interpolated using bilinear resampling to impose global consistency for the final imputed gene expression. My main question is how could gene expression data be treated like an "image"? In image data, the rows and columns are ordered. However, genes and cells in scRNA-seq data are not naturally ordered. The authors didn't provide any justifications on why the proposed "image"-based approach is appropriate. It is important to randomly change the orders of genes and cells and show that SERM's results do not depend on gene and cell orders.
2. The main goal of SERM is to impute missing gene expression. As such, when evaluating the performance of SERM, the authors randomly zeroed expression values for a certain percentage of genes. This evaluation doesn't reflect the typical situation that one would encounter in real studies. In real scRNA-seq studies, a more important task is to denoise the data and recover the unobserved true expression. Therefore, a more relevant evaluation is to downsample the original data and evaluate how SERM performs for gene expression denoising and true expression recovery.
3. Batch effects are unavoidable in large scale scRNA-seq studies. The authors should evaluate whether SERM is robust to batch effects.
4. The program is written in MATLAB, but in genomics, most people use Python or R. The authors should rewrite the program in Python if the goal is to have the method be utilized by the genomics community.

Reviewer #2 (Remarks to the Author):

This manuscript provides a new imputation method for single-cell RNA-seq. It uses deep learning model to learn an ideal distribution of data, and uses the idea of self-consistency to impute the raw data. The idea is new and interesting, but I have several comments on the manuscript.

1. The method should be compared to a range of other deep-learning-based imputation methods such as scVI and DCA.
2. The authors claim that 'However, as the deep learning techniques are purely data-driven without incorporation of any physical model, the data imputation becomes complex black-box operations and thus difficult to control. As a result, the obtained results may be distorted and lack trustworthiness.' Since the proposed method also relies on deep learning, will the proposed model face the same problem?
3. The first step of SERM is to use an autoencoder to obtain an ideal version of the data. Why not directly apply autoencoder to the whole dataset and use the result from autoencoder as the imputed values (which is similar to DCA)?
4. The autoencoder uses a subset of data as input. It is not clear from the manuscript how this subset is chosen. Is it chosen randomly?
5. In curve fitting, why not use an empirical distribution? It is questionable if the three default distributions are representative enough.
6. It is questionable if the fitted curve can be applied to all ROIs. For example, if the columns are ordered by different cell types and each cell type has different gene expression distributions.
7. It is not clear if the input data is raw count or log normalized data. If the input is

count, is the library size and other technical artifacts adjusted?

8. For Adaptive equalization of histogram, how about simply using the quantile normalization method?

9. Consider reorganizing Fig 2-4 so that the full name of the methods are directly shown on plots. It is now quite inconvenient to find the method names in the figure legends. Also it would be more clear if the subfigures are organized in a more conventional way of A,B,C rather than (a1) (a2) (b1) (b2)...

10. In Fig 4 (a3) (b3), raw data without imputation is used as gold standard. If raw data is gold standard, what's the point of imputation at the first place? A better evaluation criteria could be to calculate the correlation between inferred pseudotime with embryo stages

11. For the trajectory analysis, consider other pseudotime methods such as Monocle, Slingshot, TSCAN.

12. For clustering analysis, the authors state 'We found that none of the traditional methods was able to impute these datasets within a week on a personal computer with an Intel Core i9 processor and 64GB RAM.' This is not convincing enough. Consider doing a computational time comparison with cells subsampled from the whole data so that other methods can finish. In my experience, at least MAGIC is extremely fast and I wonder how long it takes.

13. Related to 11, the clustering performance should be evaluated for all competing methods with different numbers of subsets of cells.

Reviewer #3 (Remarks to the Author):

The paper presents a method for imputation in scRNA-Seq. This is a very relevant issue and the results presented are impressive.

The authors use deep learning and model of distributions. They impute through histogram equalization in sliding window ROIs on the gene expression data. Importantly, there is a nice comparison with other methods.

GENERAL:

Intro:

- Nice introduction. Describe imputation methods in 3 categories (sparsity probabilistic model-based, averaging/smoothing, reconstruction with deep learning or low-rank matrix assumption?) and describe contrasts

Results:

- Splatter simulator for scRNA-seq with or without dropout
- Allow computation of correlations between original data (no dropout) and imputations on dropout data using the different methods compared.
- Best correlation results with SERM
- Best visualization of simulated classes with PCA, tSNE and UMAP
- Real scRNA-seq
- 4 experiments bone marrow and leukemia, mammalian brains, mouse intestinal epithelium, engineered 3D neural tissues
- Percentage of original data filled with zeros  CONCERN! Imputation will try to recover these dropouts, but also the ones already existing in the data because of the nature of scRNA-seq data. Are comparisons in terms of correlations right or fair, then? We have to take into account that correlations are computed between the imputed data and the original data (before the insertion of zeros) that we know that is not a real reference standard because of the unknown dropouts that are not really zeros...
- Best results in terms of visualization, correlations and different measures cluster-related
- Cell trajectory analysis with PHATE
- Experiments zebrafish and human embryonic stem cells
- Best PHATE visualizations, correlations and geodesic distances across different timepoints in trajectories
- Scalability - Imputation of large datasets

- 2 datasets: Human cell landscape & mouse cell atlas
- No traditional methods can perform the imputation (PC Intel i9, 64GB RAM)
- SAUCIE and DeepImpute can do it  Why these results are not shown?? If they are worse...show them in supp! Also the cluster-based results in Figure 6
- New clusters discovered  Could they (at least one or some of them) be analyzed to check the differences with respect to the reference cell class and at least hypothesize about the identity of the potentially new (sub)celltype? In the discussion, authors claim that SERM allows the discovery of novel gene or cell types, but they do not demonstrate that they are actually newly discovered cell types.

Discussion and Others:

 Why in supplementary datasets MAGIC was the only other software used for analysis and for comparison with SERM? Makes more sense to do it with DeepImpute that seems to be the second best option after SERM in almost all datasets tested with multiple methods.

 Why the deep learning model and the histogram fitted model must be created from a fraction of all data (this point introduces problems to use SERM with small datasets, as stated by the authors in the discussion), and not the whole dataset?

 The idea of reconstructing "ideal" data using autoencoders is the same in SAUCIE. Why results are so different when reconstructing data directly (SAUCIE) instead of forcing the distributions of data to fit the distribution obtained from the reconstructions?

Methods:

 How was selected the portion/fraction of data used in each experiment for training the network? Fix number? Proportion of whole data? I guess it is a subset of cells but all the genes, right? How results change when changing the amount of data used for reconstruction and generation of distribution model?

 The proposed method "force" the same distribution in each subset (portion of cells and portion of genes) determined by ROIs generated through sliding window. Justify why the histogram distribution of all the subsets must/can be the same.

It would be valid if we have, for instance, an experiment with only one (sorted) cell type where we expect that many more dropouts can be real zeros? Another problematic example could be if cells are ordered by celltype in the expression matrix. Discuss

 Architecture and hyperparameters of Network must be provided (learning rate, number of layers (it seems to be only 3 layers: input, latent_size2, and output; but not stated), activation function, optimizer, initialization....)

 Are the parameters, hyperparameters, fraction/number of cells used for training the network, the same in all the experiments, or they were adapted depending on the dataset, the size of the dataset,...? Same question for parameters for fitting distributions

 Have the sequencing depth of the experiment any effect on the results (e.g. low depth = worse results)?

 Authors uses no overlap of sliding window. How are the results when using overlapping? Provide some results

 In all experiments performed the ROI has width and height 1/2 of total width and length of count matrix? Also in the large ones (e.g. mouse cell atlas)?

 How results change when changing the size of the ROI? A comparative study comparing a set of different sizes and also sliding distances should be performed at least in one dataset to demonstrate:

 The advantages of this divide&conquer strategy (apart from the improvement on scalability because of not need to process the whole dataset at once)

 The actual scalability to huge datasets. It seems that in all experiments 1/4 of the dataset is computed each time, which can be too much in extremely big datasets or when using computers with lower specifications.

 The decision of not using overlapping with this sliding window strategy.

Especially interesting is the case when there is actually no sliding window, but the ROI is the whole dataset (select a dataset that is not extremely big to be sure that all data can be analyzed at once)

 How results change when permuting the genes? And when permuting the cells?

Does the order matter when analyzing the different portions of data?

 PHATE configuration, parameters, etc were not described in Methods

 Clustering method (and parameters) used in cell landscape and mouse cell atlas datasets is not described in the manuscript. Authors claim that SERM allows to discover new cell types (or clusters) after imputation, but this could be just an effect of the clustering method or parameters used in each case. K-means (supervised by fixing number of desired clusters) clustering was used for computation of cluster quality indexes, but an unsupervised clustering technique must have been used to get different number of clusters in those two datasets that support the idea of potential discovery of new cell types.

Figs:

 Fig1: improve and label steps in figures/plots for consistency with the text.

 Fig2: Bins should be the same in histograms of all cases, for a proper comparison.

Add names of methods also in figure and not only in caption (needed in most of figures, actually).

 Fig5 & S11: Show number representing the reference standard (celltypes) accompanying the colors, to be able to compare same population in original data and imputed data (colors are too similar and the identification of a specific cell type is not possible in these figures). In these figures at these moment numbers represent cluster while the color is the ref standard. I suggest to duplicate the plots, one colored and numbered by ref std and other colored and numbered by cluster, displayed one next to the other.

Supp Figs and Material:

 FigS1: (a)-(d) missing in figure

 FigS20: legend is also for FigS21.

 Section 3 to 6: Add color legends for celltypes of datasets, same as in section 7

Reviewer #1 (Remarks to the Author)

Islam et al. developed SERM, a deep learning method that aims to impute gene expression in single-cell RNA-seq. Numerous methods have been developed to denoise or impute gene expression in scRNA-seq. The advantage of SERM compared to other methods is its speed. The authors showed that it has at least 100-fold increase in computational efficiency as compared to other methods. I have several major concerns about the paper.

1. The key idea of the paper is to consider gene expression matrix as an “image” and focus on gene expression imputation within each Region of Interest (ROI) and then slide the ROI

along x and y direction throughout the entire gene expression matrix. Within each ROI, gene expression imputation is performed by histogram equalization and all ROIs are then interpolated using bilinear resampling to impose global consistency for the final imputed gene expression. My main question is how could gene expression data be treated like an “image”? In image data, the rows and columns are ordered. However, genes and cells in scRNA-seq data are not naturally ordered. The authors didn’t provide any justifications on why the proposed “image”-based approach is appropriate. It is important to randomly change the orders of genes and cells and show that SERM’s results do not depend on gene and cell orders.

Response:

Arrangement of the gene expression matrix as an image with ordered data elements is not necessary for histogram equalization and gene imputation. An implicit assumption made in our formulation is that the coverage of ROI is sufficiently large so that the captured data distribution is representative of that across the entire matrix. While this assumption is generally true, there may exist special situations where the data in an ROI(s) may not be sufficiently representative of the entire data, as rightfully pointed out by the reviewers. In this resubmission, this issue is addressed by adding an optional step, in which the positions of cells or genes can be randomized through column/row permutations to ensure unbiased distribution in different parts of the matrix. We note that in the majority of applications, it is not necessary to take the optional data randomization step as the genomic data are generally not ordered. To illustrate this, we computed the Pearson correlation values of imputed and reference data for more than twenty expression datasets with 1000 different random alterations of cells and genes (Fig. S22 shows an example for one of the datasets). We found that the correlation values remain almost identical for all the alterations (see Fig. S22 for an example), showing that SERM is robust against the random changes in the positions of the genes and cells. The following paragraph has been added to the revised discussion section:

“In SERM, a data-driven histogram equalization is performed on each ROI to impute the gene expression values. Although histogram equalization is a popular technique in image processing for contrast enhancement, the assumption of gene expression matrix as an image with ordered data elements is not necessary here. An implicit assumption made in our formulation is that the coverage of ROI is sufficiently large so that the captured data distribution in the ROI is representative of the entire matrix. As only the distribution of expression values in a sufficiently large ROI is considered, the order of rows and columns is generally not important. In other words, if a gene (column) or a cell (row) is interchanged with another one, the distribution of the gene expression values, as well as the resultant imputation, should not change much. To ensure unbiased data distribution in different parts of the matrix, an optional step in which the columns/rows can be permuted randomly is included in our implementation. However, we note that in most cases, this randomization step is not necessary as the genomic data are generally not ordered. To illustrate this, we computed the correlation values from SERM imputed data for more than twenty expression datasets with 1000 different random alterations of cells and genes (Fig.S22 shows an example for one of the datasets). We found that the correlation values remain

almost identical (Fig.S22), showing that SERM is robust against random changes in the positions of the genes and cells.”

2. The main goal of SERM is to impute missing gene expression. As such, when evaluating the performance of SERM, the authors randomly zeroed expression values for a certain percentage of genes. This evaluation doesn't reflect the typical situation that one would encounter in real studies. In real scRNA-seq studies, a more important task is to denoise the data and recover the unobserved true expression. Therefore, a more relevant evaluation is to downsample the original data and evaluate how SERM performs for gene expression denoising and true expression recovery.

Response: We want to thank the reviewer for this insightful suggestion. The imputation results for the down-sampled data by different techniques are added to the manuscript. The added data show clearly the superiority of SERM over the existing techniques.

3. Batch effects are unavoidable in large scale scRNA-seq studies. The authors should evaluate whether SERM is robust to batch effects.

Response: We have added the following paragraph in the revised discussion:

“Although batch effects are not considered explicitly in the formulation of SERM, the technique is generally less susceptible to data obtained by using different technologies or protocols (see Fig. S28, in which results from different data acquisition techniques with batch effects are shown). The reason behind this is that SERM enforces a single learned distribution to all the data from different batches. Assuming the data from different batches have different distributions, SERM forces their distributions to follow the distribution it learned, alleviating any potential bias and artifacts. Nonetheless, we would suggest using SERM on the data from different batches separately and then employing a dedicated batch-effect correction technique (such as Seurat ¹).”

4. The program is written in MATLAB, but in genomics, most people use Python or R. The authors should rewrite the program in Python if the goal is to have the method be utilized by the genomics community.

Response: We have added Python implementation of SERM and shared our codes via Github (<https://github.com/xinglab-ai/self-consistent-expression-recovery-machine>). Our Python codes are also shared as a Code Ocean capsule so that the reviewers can reproduce the SERM results with a single click.

Reviewer #2 (Remarks to the Author)

This manuscript provides a new imputation method for single-cell RNA-seq. It uses deep learning model to learn an ideal distribution of data, and uses the idea of self-consistency to

impute the raw data. The idea is new and interesting, but I have several comments on the manuscript.

1. The method should be compared to a range of other deep-learning-based imputation methods such as scVI and DCA.

Response: We have compared our technique with scVI and DCA in the revised manuscript. In comparison to both these techniques, SERM performs better as found in our analyses.

2. The authors claim that ‘However, as the deep learning techniques are purely data-driven without incorporation of any physical model, the data imputation becomes complex black-box operations and thus difficult to control. As a result, the obtained results may be distorted and lack trustworthiness.’ Since the proposed method also relies on deep learning, will the proposed model face the same problem?

Response: We have added the following paragraph in the discussion section of the revised manuscript.

“As discussed in the introduction, a deep learning model generally proceeds in a black-box fashion with little control given to the user. In SERM, only the data distribution is learned via deep learning and the actual imputation is performed by using an analytical histogram matching technique. Thus, the method is more transparent than traditional deep learning-based imputation techniques. We note that, in SERM, the distribution can also be preset by the user, which makes it easy for a user to investigate the effect of the learned distribution on the imputation. This also empowers the user to examine the results in depth when a question arises about the modeling, which not possible in other deep learning-based imputation techniques.”

3. The first step of SERM is to use an autoencoder to obtain an ideal version of the data. Why not directly apply autoencoder to the whole dataset and use the result from autoencoder as the imputed values (which is similar to DCA)?

Response: We have added the following paragraph in the discussion section of the revised manuscript.

“In DCA, SAUCIE or other encoder-based imputations, the objective function is based on reconstructing the original data from the latent representation. The approach affords a unique way to denoise the data. However, the methods do not put any constraint on the distribution of the data. By leveraging an analytical distribution to impute the values, SERM not only denoises the data but also imposes a natural guidance on the data distribution, leading to substantially improved imputation accuracy and efficiency. While it may be possible to create an autoencoder with constraints on the data distribution, computationally, the approach would be much less efficient than SERM, especially when the data size increases.”

4. The autoencoder uses a subset of data as input. It is not clear from the manuscript how this subset is chosen. Is it chosen randomly?

Response: The subset of the data was chosen randomly. We made it clear in the revised manuscript (please see methods section (“Implementation and parameter settings”) for details).

0. In curve fitting, why not use an empirical distribution? It is questionable if the three default distributions are representative enough.

Response: We have added the following paragraph in the discussion section of the revised manuscript.

“In SERM, we have considered three analytical distributions (Gaussian, Rayleigh, and exponential) to model the data. Other distributions can also be included in SERM (see Python/Matlab codes of SERM). One can also use empirical distribution to characterize the data. In Fig.S21, we have included SERM-imputed results for one dataset, when the empirical distribution is used in place of the analytical distributions. It is seen that the SERM with an empirical distribution performs inferiorly to that with learned analytical distributions. For ultra large data (e.g., on a scale of millions), the empirical distribution may characterize the data better and lead to improved performance.”

1. It is questionable if the fitted curve can be applied to all ROIs. For example, if the columns are ordered by different cell types and each cell type has different gene expression distributions.

Response: Thanks for bringing this important issue up. An implicit assumption made in our formulation is that the coverage of ROI is sufficiently large so that the captured data distribution is representative of that across the entire matrix. While this assumption is generally true, there may exist special situations where the data in an ROI(s) may not be sufficiently representative of the entire data, as correctly pointed out by the reviewers. In this resubmission, this issue is addressed by adding an optional step, in which the positions of cells or genes can be randomized through column/row permutations to ensure unbiased distribution in different parts of the matrix. We note that, in the majority of applications, it is not necessary to take the optional data randomization step as the genomic data are generally not ordered. To illustrate this, we computed the Pearson correlation values of imputed and reference data for more than twenty expression datasets with 1000 different random alterations of cells and genes (Fig. S22 shows an example for one of the dataset). We found that the correlation values remain almost identical (see Fig. S22 for an example), showing that SERM is robust against random changes in the positions of the genes and cells. The following paragraph has been added to the revised discussion section:

“In SERM, a data-driven histogram equalization is performed on each ROI to impute the gene expression values. Although histogram equalization is a popular technique in image processing for contrast enhancement, the assumption of gene expression matrix as an image with ordered data elements is not necessary here. An implicit assumption made in our formulation is that the coverage of ROI is sufficiently large so that the captured data distribution in the ROI is representative of the entire matrix. As only the distribution of expression values in a sufficiently large ROI is considered, the order of rows and columns is generally not important. In other words, if a gene (column) or a cell (row) is interchanged with another one, the distribution of the gene

expression values, as well as the resultant imputation, should not change much. To ensure unbiased data distribution in different parts of the matrix, an optional step in which the columns/rows can be permuted randomly is included in our implementation. However, we note that in most cases, this randomization step is not necessary as the genomic data are generally not ordered. To illustrate this, we computed the correlation values from SERM imputed data for more than twenty expression datasets with 1000 different random alterations of cells and genes (Fig.S22 shows an example for one of the datasets). We found that the correlation values remain almost identical (Fig.S22), showing that SERM is robust against random changes in the positions of the genes and cells.”

To demonstrate the performance of SERM for the situation that the reviewer referred to, we have added analysis on a dataset, where the columns are ordered by different cell types. The performance of SERM in this case is added in Fig. S18. It is seen that SERM’s performance degrades as compared to the original “un-ordered” data by only less than 1%. The reason behind this is that, while the distribution of gene expression changes from cell type to cell type, the change is not very significant (see Fig. S20, where we show the gene expression distribution for 10 different cell types of one dataset). It is seen from Fig. S20 that, in all cell types, the distribution of gene expression follows a bell-shaped curve with slight differences in variance and mean.

5. It is not clear if the input data is raw count or log normalized data. If the input is count, is the library size and other technical artifacts adjusted?

Response: The input data can be either raw count or library size normalized data (Please see Fig. S17, where we show that SERM is equally applicable for data with and without library normalization). Although SERM is generally robust to batch effect and technical artifacts (shown in Fig. S28), we would advise users to take precautions to avoid any known technical artifacts as it may change the data distribution and SERM’s performance. We have added the following paragraph in the revised discussion:

“Although batch effects are not considered explicitly in the formulation of SERM, the technique is generally less susceptible to data obtained by using different technologies or protocols (see Fig. S28, in which results from different data acquisition techniques with batch effects are shown). The reason behind this is that SERM enforces a single learned distribution to all the data from different batches. Assuming the data from different batches have different distributions, SERM forces their distributions to follow the distribution it learned, alleviating any potential bias and artifacts. Nonetheless, we would suggest using SERM on the data from different batches separately and then employing a dedicated batch-effect correction technique (such as Seurat ¹).”

6. For Adaptive equalization of histogram, how about simply using the quantile normalization method?

Response: We show SERM results with quantile normalization in Fig. S16, where it is seen that the SERM performance is degraded. This is likely because of the robust and superior performance of adaptive histogram equalization as compared to quantile equalization ².

9. Consider reorganizing Fig 2-4 so that the full name of the methods are directly shown on plots. It is now quite inconvenient to find the method names in the figure legends. Also it would be more clear if the subfigures are organized in a more conventional way of A,B,C rather than (a1) (a2) (b1) (b2)...

Response: We have revised our figures based on the reviewer's suggestion.

10. In Fig 4 (a3) (b3), raw data without imputation is used as gold standard. If raw data is gold standard, what's the point of imputation at the first place? A better evaluation criteria could be to calculate the correlation between inferred pseudotime with embryo stages

Response: Thanks for the insightful suggestions and we have implemented the suggested benchmarking method. We have added the correlation value of inferred pseudotime by monocle³ from imputed expression values by different methods with embryo stages in the revised manuscript (Fig. 5), where we see again that SERM provides the best correlation values for all the datasets.

11. For the trajectory analysis, consider other pseudotime methods such as Monocle, Slingshot, TSCAN.

Response: We have added trajectory analysis results for monocle, Slingshot, and TSCAN in the revised manuscript and supplementary (Figs. 5, S35, S36). These methods with SERM-imputed data perform better than those obtained by using input data imputed with other imputation methods.

12. For clustering analysis, the authors state 'We found that none of the traditional methods was able to impute these datasets within a week on a personal computer with an Intel Core i9 processor and 64GB RAM.' This is not convincing enough. Consider doing a computational time comparison with cells subsampled from the whole data so that other methods can finish. In my experience, at least MAGIC is extremely fast and I wonder how long it takes.

Response: We have added results for all the techniques on subsampled data (by a factor of 20) in the revised manuscript. We have reported the performance and computational time in the revised manuscript and supplementary (Figs. 7-9, S34), where it is seen that SERM performs much better with a substantial gain in computational efficiency.

13. Related to 11, the clustering performance should be evaluated for all competing methods with different numbers of subsets of cells.

Response: We have added quantitative and qualitative results of different techniques for under sampled data (by a factor of 20-Figs.7-9) and the full data (Figs.S29-S34). It is seen from these figures that SERM performs significantly better than all other competing techniques.

Reviewer #3 (Remarks to the Author)

The paper presents a method for imputation in scRNA-Seq. This is a very relevant issue and the results presented are impressive.

The authors use deep learning and model of distributions. They impute through histogram equalization in sliding window ROIs on the gene expression data.

Importantly, there is a nice comparison with other methods.

GENERAL:

Intro:

- Nice introduction. Describe imputation methods in 3 categories (sparsity probabilistic model-based, averaging/smoothing, reconstruction with deep learning or low-rank matrix assumption?) and describe contrast

Response: We want to thank the reviewer for his nice summary of our introduction section.

Results:

- Splatter simulator for scRNA-seq with or without dropout, - Allow computation of correlations between original data (no dropout) and imputations on dropout data using the different methods compared. - Best correlation results with SERM,- Best visualization of simulated classes with PCA, tSNE and UMAP,- Real scRNA-seq,- 4 experiments bone marrow and leukemia, mammalian brains, mouse intestinal epithelium, engineered 3D neural tissues

Response: We are grateful for the insightful summary of our methods and analyses.

- Percentage of original data filled with zeros  CONCERN! Imputation will try to recover these dropouts, but also the ones already existing in the data because of the nature of scRNA-seq data. Are comparisons in terms of correlations right or fair, then? We have to take into account that correlations are computed between the imputed data and the original data (before the insertion of zeros) that we know that is not a real reference standard because of the unknown dropouts that are not really zeros...

Response: We have added the results for imputing the downsampled data by different techniques. We note that the utilized downsampling procedure is a standard technique for evaluating the imputation techniques (Refs. ^{4,5}).The added data (Figs. 1-6) show clearly that SERM significantly outperforms the existing techniques.

- Best results in terms of visualization, correlations and different measures cluster-related

Response: The reviewer is correct that SERM provides the best results in terms of visualization, correlations, and different cluster-related measures.

- Cell trajectory analysis with PHATE, - Experiments zebrafish and human embryonic stem cells, - Best PHATE visualizations, correlations and geodesic distances across different timepoints in trajectories, - Scalability - Imputation of large datasets, - 2 datasets: Human cell landscape & mouse cell atlas, - No traditional methods can perform the imputation (PC Intel i9, 64GB RAM)

Response: We want to thank the reviewer for the insightful summary of our analyses.

- SAUCIE and DeepImpute can do it  Why these results are not shown?? If they are worse...show them in supp! Also the cluster-based results in Figure 6

Response: Based on the reviewer's suggestion, SAUCIE and DeepImpute results are included in the supplementary (Figs. S31, S32).

- New clusters discovered  Could they (at least one or some of them) be analyzed to check the differences with respect to the reference cell class and at least hypothesize about the identity of the potentially new (sub)celltype? In the discussion, authors claim that SERM allows the discovery of novel gene or cell types, but they do not demonstrate that they are actually newly discovered cell types.

Response: In the analyzed datasets, there were no unrecognized cells. Thus, we could not claim the recognition of unknown cells from SERM-imputed data. We have edited our claim from "SERM allows the discovery of novel gene or cell types" to "In the visualizations of projected data in a low dimension, many new cell clusters are found after SERM-imputation. Thus, using SERM, it may be possible to discover novel biological information such as cell or gene types from very large scRNA-seq data."

Discussion and Others:

 Why in supplementary datasets MAGIC was the only other software used for analysis and for comparison with SERM? Makes more sense to do it with DeepImpute that seems to be the second best option after SERM in almost all datasets tested with multiple methods.

Response: We have added DeepImpute results for all the supplementary datasets.

 Why the deep learning model and the histogram fitted model must be created from a fraction of all data (this point introduces problems to use SERM with small datasets, as stated by the authors in the discussion), and not the whole dataset?

Response: The models can also be created from the whole dataset in SERM (see added Fig. S26, where we show that this does not change the performance of SERM). However, to reduce the computational burden, we use a subset of the data in SERM for learning the denoised data distribution. We have made this point clear in this resubmission (see "Implementation and parameter settings").

 The idea of reconstructing "ideal" data using autoencoders is the same in SAUCIE. Why results are so different when reconstructing data directly (SAUCIE) instead of forcing the distributions of data to fit the distribution obtained from the reconstructions?

Response: We have added the following paragraph in the discussion section of the revised manuscript.

"In DCA, SAUCIE or other encoder-based imputations, the objective function is based on reconstructing the original data from the latent representation. The approach affords a unique way to denoise the data. However, the methods do not put any constraint on the distribution of the data. By leveraging an analytical distribution to impute the values, SERM not only denoises the data but also imposes a natural guidance on the data distribution, leading to substantially improved imputation accuracy and efficiency. While it may be possible to create an autoencoder with constraints on the data distribution, computationally, the approach would be much less efficient than SERM, especially when the data size increases."

Methods:

 How was selected the portion/fraction of data used in each experiment for training the network? Fix number? Proportion of whole data? I guess it is a subset of cells but all the genes, right? How results change when changing the amount of data used for reconstruction and generation of distribution model?

Response: We selected randomly 2000 data points. Yes, all the genes are selected. We have made it clear in the revised manuscript (see "Implementation and parameter settings"). In supplementary section 8, we have added SERM results for data of 43 different sizes (at increment of 100 data points) used for reconstruction and generation of distribution model for 3D neural tissue dataset. It is seen that when the data is small (≤ 1800), the learned distribution in SERM and its performance changes for different number of data points. But for more than 1800 data points, the learned distribution and performance remain almost same.

 The proposed method "force" the same distribution in each subset (portion of cells and portion of genes) determined by ROIs generated through sliding window. Justify why the histogram distribution of all the subsets must/can be the same.

Response: Thanks for bringing this important issue up. An implicit assumption made in our formulation is that the coverage of ROI is sufficiently large so that the captured data distribution is representative of that across the entire matrix. While this assumption is generally true, there may exist special situations where the data in an ROI(s) may not be sufficiently representative of the entire data, as correctly pointed out by the reviewers. In this resubmission, this issue is addressed by adding an optional step, in which the positions of cells or genes can be randomized through column/row permutations to ensure unbiased distribution in different parts of the matrix. We note that, in the majority of applications, it is not necessary to take the optional data randomization step as the genomic data are generally not ordered. To illustrate this, we computed

the Pearson correlation values of imputed and reference data for more than twenty expression datasets with 1000 different random alterations of cells and genes (Fig. S22 shows an example for one of the dataset). We found that the correlation values remain almost identical (see Fig. S22 for an example), showing that SERM is robust against random changes in the positions of the genes and cells. The following paragraph has been added to the revised discussion section:

“In SERM, a data-driven histogram equalization is performed on each ROI to impute the gene expression values. Although histogram equalization is a popular technique in image processing for contrast enhancement, the assumption of gene expression matrix as an image with ordered data elements is not necessary here. An implicit assumption made in our formulation is that the coverage of ROI is sufficiently large so that the captured data distribution in the ROI is representative of the entire matrix. As only the distribution of expression values in a sufficiently large ROI is considered, the order of rows and columns is generally not important. In other words, if a gene (column) or a cell (row) is interchanged with another one, the distribution of the gene expression values, as well as the resultant imputation, should not change much. To ensure unbiased data distribution in different parts of the matrix, an optional step in which the columns/rows can be permuted randomly is included in our implementation. However, we note that in most cases, this randomization step is not necessary as the genomic data are generally not ordered. To illustrate this, we computed the correlation values from SERM imputed data for more than twenty expression datasets with 1000 different random alterations of cells and genes (Fig.S22 shows an example for one of the datasets). We found that the correlation values remain almost identical (Fig.S22), showing that SERM is robust against random changes in the positions of the genes and cells.”

We have added the distribution of the data in different ROIs from simulated dropout-free gene expression data in Fig.S1, where it is seen that the distribution remains almost same in different ROIs of the expression matrix.

It would be valid if we have, for instance, an experiment with only one (sorted) cell type where we expect that many more dropouts can be real zeros? Another problematic example could be if cells are ordered by celltype in the expression matrix. Discuss

Response: To demonstrate the performance of SERM for the situations that the reviewer referred to, we have added two analyses on the cellular taxonomy dataset, where (1) only data of one cell type is imputed using SERM and compared with the result of the full data, (2) the data are ordered by different cell types. The performances of SERM in these cases are added in Fig. S18 and S19. It is seen that SERM's performance degrades as compared to the original data by only less than 2% for these cases. The reason behind this is that, while the distribution of gene expression changes from cell type to cell type, the change is not very significant (see Fig. S20, where we show the gene expression distribution for 10 different cell types of one dataset). It is seen from Fig. S20 that, in all cell types, the distribution of gene expression has a similar curve with slight differences in variance and mean.

We have added the following sentences in the revised discussion:

“We have also showed examples where the dataset is arranged based on cell-type or only one type of cell is considered (Figs. S18 and S19). However, even in these special cases, the accuracy of SERM drops by only less than 2% and the method can still be employed for high-performance gene imputation. The reason behind this is that, while the distribution of gene expression changes from cell type to cell type, the change is not very significant (see Fig. S20, where we show the gene expression distribution for 10 different cell types of a dataset). It is seen from Fig.S20 that, in all cell types, the distribution of gene expression has a similar curve with slight differences in variance and mean.”

 Architecture and hyperparameters of Network must be provided (learning rate, number of layers (it seems to be only 3 layers: input, latent_size2, and output; but not stated), activation function, optimizer, initialization....)

Response: We have added details of the hyperparameters of the network in the revised manuscript (methods section). See “Implementation and parameter settings” subsection.

 Are the parameters, hyperparameters, fraction/number of cells used for training the network, the same in all the experiments, or they were adapted depending on the dataset, the size of the dataset,...? Same question for parameters for fitting distributions

Response: Yes, the parameters are the same in all experiments. We have made this clear in the revised manuscript. See “Implementation and parameter settings” subsection.

 Have the sequencing depth of the experiment any effect on the results (e.g. low depth = worse results)?

Response: For low depth sequencing the performance degrades for all the methods including SERM. However, SERM performs substantially better than others for all sequencing depths (see Figs.3 and 4).

 Authors uses no overlap of sliding window. How are the results when using overlapping? Provide some results

Response: We have added results with different overlapping windows in the revised supplementary (Fig.S27) for 3D neural tissue dataset. SERM with overlapping windows performs slightly better than SERM without overlapped window (performance difference normally varies by 1%). However, SERM without overlapping windows is much faster than SERM with overlapping windows.

 In all experiments performed the ROI has width and heigh 1/2 of total width and length of count matrix? Also in the large ones (e.g. mouse cell atlas)?

Response: We have divided the human atlas data into 48 and mouse atlas data into 28 ROIs - this has been made clear in the revised manuscript. For all other datasets, we used $1/2$ of the width and length. We note that the ROI size depends on the total size of the data. For

very large datasets such as mouse atlas, SERM needs a higher number of ROI. That said, SERM is in general robust against the number of ROIs and it provides high quality imputation for a wide range of ROI numbers (demonstrated in Fig. S26).

 How results change when changing the size of the ROI? A comparative study comparing a set of different sizes and also sliding distances should be performed at least in one dataset to demonstrate:

Response: We have added results for different ROI sizes and sliding distances (overlap percentage) in the revised supplementary (Figs. S26 and S27). It is seen that SERM performs well for a large range of the parameters, which proves its robustness to parameter selection.

 The advantages of this divide&conquer strategy (apart from the improvement on scalability because of not need to process the whole dataset at once)

Response: We have added the following paragraph in the revised discussion section:

“Apart from the computational efficiency and scalability, SERM substantially improves the performance of the imputation process with augmented interpretability in comparison to the existing deep learning techniques.”

Please also see the seventh paragraph of the discussion section.

 The actual scalability to huge datasets. It seems that in all experiments 1/4 of the dataset is computed each time, which can be too much in extremely big datasets or when using computers with lower specifications.

Response: SERM is scalable to datasets of all sizes. In our analyses, we have divided the human cell atlas and mouse atlas data into 48 and 28 ROIs. For all other datasets which are relatively small, we used $1/2$ of the width and length. We note that the ROI size depends on the total size of the data. For very large datasets such as mouse cell atlas, SERM needs a higher number of ROI. As the selected ROIs are normally small in size (<12000 cells, <2000 genes), SERM is executable even in PC with lower specifications. We found that, SERM is in general robust against the number of ROIs and it provides high quality imputation for a wide range of ROI numbers (demonstrated in Fig. S26).

 The decision of not using overlapping with this sliding window strategy.

Response: We have not used the overlapping windows to reduce the computational burden. In this resubmission, we have added results for the overlapping windows (Fig. S26). SERM with the overlapping window performs slightly better than SERM without overlapped window (performance difference normally varies by 1%). However, SERM without overlapping windows is at least 4 times faster than SERM with overlapping windows (for 25% overlapping).

Epecially interesting is the case when there is actually no sliding window, but the ROI is the whole dataset (select a dataset that is not extremely big to be sure that all data can be analyzed at once)

Response: When the ROI is large, SERM still would be able to recover the ideal expressions to certain extent. However, the performance may not be optimum as histogram equalization on a large dataset may not be accurate. As an example, we have added results for different ROI sizes (from 1/40th fraction to full dataset) in Fig.S26. It is seen that SERM's performance remains almost unchanged for different sizes of ROI.

 How results change when permuting the genes? And when permuting the cells? Does the order matter when analyzing the different portions of data?

Response: We have added the following paragraph in the revised discussion section:

“In SERM, a data-driven histogram equalization is performed on each ROI to impute the gene expression values. Although histogram equalization is a popular technique in image processing for contrast enhancement, the assumption of gene expression matrix as an image with ordered data elements is not necessary here. An implicit assumption made in our formulation is that the coverage of ROI is sufficiently large so that the captured data distribution in the ROI is representative of the entire matrix. As only the distribution of expression values in a sufficiently large ROI is considered, the order of rows and columns is generally not important. In other words, if a gene (column) or a cell (row) is interchanged with another one, the distribution of the gene expression values, as well as the resultant imputation, should not change much. To ensure unbiased data distribution in different parts of the matrix, an optional step in which the columns/rows can be permuted randomly is included in our implementation. However, we note that in most cases, this randomization step is not necessary as the genomic data are generally not ordered. To illustrate this, we computed the correlation values from SERM imputed data for more than twenty expression datasets with 1000 different random alterations of cells and genes (Fig.S22 shows an example for one of the datasets). We found that the correlation values remain almost identical (Fig.S22), showing that SERM is robust against random changes in the positions of the genes and cells.”

 PHATE configuration, parameters, etc were not described in Methods

Response: We have added the details of PHATE configurations and parameters. See “Implementation and parameter settings” in the methods section.

 Clustering method (and parameters) used in cell landscape and mouse cell atlas datasets is not described in the manuscript. Authors claim that SERM allows to discover new cell types (or clusters) after imputation, but this could be just an effect of the clustering method or parameters used in each case. K-means (supervised by fixing number of desired clusters) clustering was used for computation of cluster quality indexes, but an unsupervised clustering

technique must have been used to get different number of clusters in those two datasets that support the idea of potential discovery of new cell types.

Response: We have added the details of the parameters and settings of the k-means clustering in the revised manuscript. See “Computation of NMI, accuracy and cluster quality indices” subsection in the methods section. We ran k-means clustering 1000 times and reported the average clustering accuracy and cluster quality indices in the manuscript.

In the analyzed datasets, there were no unrecognized cells. Thus, we could not claim the recognition of unknown cells from SERM-imputed data. We have edited our claim from “SERM allows the discovery of novel gene or cell types” to “In the visualizations of projected data in a low dimension, many new cell clusters are found after SERM-imputation. Thus, using SERM, it may be possible to discover novel biological information such as cell or gene types from very large scRNA-seq data.”

Figs:

 Fig1: improve and label steps in figures/plots for consistency with the text.

Response: We have labeled steps in figures/plots for consistency with the text.

 Fig2: Bins should be the same in histograms of all cases, for a proper comparison. Add names of methods also in figure and not only in caption (needed in most of figures, actually).

Response: We have revised the bin size in all figures for proper comparison. We rescaled the data at first and created histograms by setting number of bins equal to 100. We also have added the method name in the figure.

 Fig5 & S11: Show number representing the reference standard (celltypes) accompanying the colors, to be able to compare same population in original data and imputed data (colors are too similar and the identification of a specific cell type is not possible in these figures). In these figures at these moment numbers represent cluster while the color is the ref standard. I suggest to duplicate the plots, one colored and numbered by ref std and other colored and numbered by cluster, displayed one next to the other.

Response: To facilitate the understanding of SERM's superior performance in finding better clusters from HCL and MCA datasets, we have added quantitative indices (clustering accuracy and cluster quality indices) in the revised manuscript for both subsampled (by a factor of 20) and full datasets. Please see Figs.7-9 and Figs.S29-S34.

Supp Figs and Material:

 FigS1: (a)-(d) missing in figure

Response: We have corrected this issue.

 FigS20: legend is also for FigS21.

Response: We have corrected this issue.

 Section 3 to 6: Add color legends for celltypes of datasets, same as in section 7

Response: We have added the color legend in all the figures.

Reference:

1. Stuart, T. *et al.* Comprehensive Integration of Single-Cell Data. *Cell* **177**, 1888-1902.e21 (2019).
2. Contrast-limited adaptive histogram equalization (CLAHE) - MATLAB `adapthisteq`. <https://www.mathworks.com/help/images/ref/adapthisteq.html>.
3. Trapnell, C. *et al.* The dynamics and regulators of cell fate decisions are revealed by pseudotemporal ordering of single cells. *Nat Biotechnol* **32**, 381–386 (2014).
4. Huang, M. *et al.* SAVER: Gene expression recovery for single-cell RNA sequencing. *Nature Methods* **15**, 539–542 (2018).
5. Miao, Z., Li, J. & Zhang, X. *scRecover: Discriminating true and false zeros in single-cell RNA-seq data for imputation*. 665323 <https://www.biorxiv.org/content/10.1101/665323v1> (2019) doi:10.1101/665323.

Reviewer #1 (Remarks to the Author):

1. I still have questions for the ROI step in the algorithm. In the revision, the authors explained that " ... this issue is addressed by adding an optional step, in which the positions of cells or genes can be randomized through column/row permutations to ensure unbiased distribution in different parts of the matrix." While I understand this might be true if all of the cells are collected from the same batch, I am not sure how this performs when the cells are collected from different batches. For the latter case, how do you permute the cells? Do you need to do any batch effects removal before doing the permutations?

2. Regarding my comment on batch effects, the authors responded that "Although batch effects are not considered explicitly in the formulation of SERM, the technique is generally less susceptible to data obtained by using different technologies or protocols (see Fig. S28, in which results from different data acquisition techniques with batch effects are shown).", but Fig. S28 is not addressing my comment on batch effects. This figure only shows the performance of different imputation methods on two different datasets. I would like to see the authors appropriately evaluate the performance of SERM when batch effects are present.

3. Regarding my comment on batch effects, the authors further responded that "Nonetheless, we would suggest using SERM on the data from different batches separately and then employing a dedicated batch-effect correction technique (such as Seurat 1)." The advantage of SERM is substantially reduced if it has to rely on a separate program such as Seurat to deal with batch effects. Most users would prefer to use a single program to conduct data analysis. Having to use the combination of SERM + another program (e.g. Seurat) is somewhat cumbersome for users.

Reviewer #2 (Remarks to the Author):

The reviewers have addressed all my concerns. I do not have further comments.

Reviewer #3 (Remarks to the Author):

Authors change their strategy by using downsampled data instead of the zero-filled strategy to generate dropout-affected data to evaluate their imputation method (and others). It is not clear if this new "downsampled data" is really valid for evaluation, and for supporting their claims.

- The process to generate the dropout-affected data is not explained. It is a critical point since all conclusions derived from the use of data in this way.
- It is not clear what are the "Original data" and the "Downsampled data" they use in their experiments and figures. It is not even clear if they perform imputation over the downsampled data and compare the results with the original data (which seems to be the case).
- Downsampled data is usually referred to extract a portion of data, which in this case would be a subset of cells and a subset of genes.
- Using this definition I do not see how imputation accuracy can be evaluated.
- If authors mean a different concept, please use different terms and provide detailed information about the construction of these "downsampled datasets" and how it allows a proper evaluation of imputation methods.

I needed to go deep into the references and code the authors provided trying to understand all these things:

- References and explanation of dropout-affected is not explained. Two references are provided within the text (18,37) and at least one of them is wrong (37).

[18. Miao, Z., Li, J. & Zhang, X. scRecover: Discriminating true and false zeros in single-cell RNA-seq data for imputation. 665323. URL <https://www.biorxiv.org/content/10.1101/665323v1>. DOI 10.1101/665323.]

[37. Abdelaal, T. et al. A comparison of automatic cell identification methods for single-cell RNA sequencing data. 20, 194. URL <https://doi.org/10.1186/s13059-019-1795-z>. DOI 10.1186/s13059-019-1795-z.]

In response to reviewers they include 2 references [4,5] (4 could be right this time):

[4. Huang, M. et al. SAVER: Gene expression recovery for single-cell RNA sequencing. Nature Methods 15, 539–542 (2018).]

[5. Miao, Z., Li, J. & Zhang, X. scRecover: Discriminating true and false zeros in single-cell RNA-seq data for imputation. 665323 <https://www.biorxiv.org/content/10.1101/665323v1> (2019) doi:10.1101/665323.]

In demo software provided by authors (<https://codeocean.com/capsule/0864915>) they only referenced Miao et al. for the generation of downsampled data, specifying the use of "countsSampling" function. This function basically reduces the number of counts in the sample. This may introduce some "simulated zeros" when the reduction is high, because of the integer nature of counts. If that is simply the way to create the downsampled data, it would have been very easy to explain and detail within the paper... (In both references there are other steps that may or may not be part of the analytical pipeline in this paper).

Note: this paper (Miao et al.) is not published in a peer-review journal, and it is just a preprint. The methodology used could be questionable.

- In Huang et al. authors generate "Reference datasets" by keeping only a portion of genes and cells (i.e. downsampled data) that they selected as high quality (genes where low amount of cells have zero expression, and cells with enough library size). From this reference data, they generated their "observed data" by simulating efficiency loss that "introduces zeros". They use these "observed data" as input for imputation, and compare results with the reference (downsampled) dataset.

By checking the codes and toy data of the paper under review:

- "Original data" used by authors during the paper, seem to be something like "reference data" in huang et al. The example provided with the code (cellular taxonomy of the bone marrow stroma) only has 1093 genes and 23092 cells, so cells and genes seem to be filtered from the actual original data (with much more cells and genes). None of this preprocessing, filtering, etc. is named or explained in the paper and it is critical. Moreover, the term "original data" is confusing if it refers to processed data or a subset of data. Please, add number of cells/genes for each dataset in their corresponding places in Methods section.

- "Downsampled data" used by authors during the paper, seem to be something like "observed data" in huang et al. The example provided with the code (cellular taxonomy of the bone marrow stroma) shows a very similar count matrix with respect to the "original" (reference), but with lower values.

- Authors seem to impute "observed data" (their "downsampled data") and compare the results with the "reference data" (their "original data"). It need to be clear in the manuscript. Next points of this review are based on that assumption.

- I have serious doubts about the validity of their "downsampled data" to evaluate imputation. Based on the example they provide with the code (<https://codeocean.com/capsule/0864915>, cellular taxonomy of the bone marrow stroma):

- The percentage of zeros is very similar in reference (51.8%) and observed (downsampled to 1%, used for imputation) data (52.47%).

- The pearson correlation between the reference and downsampled data, prior to imputation, is already very high (0.89 for all genes together, and mean=0.85 median=0.87 when computing correlations for each gene independently). It is not clear from figure 4 first row, but those values seem to be higher than the ones obtained by any of the imputation methods tested (when comparing at 1%).

In any case, correlations of unimputed data need to be added in these plots (Figure 4, Figure 5c, Figure 6,...).

- TSNE/UMAP plots do not reflect an improvement when performing imputation. Results with SERM are almost identical to the ones obtained with the "downsampled data" (no imputation). See figures 2a, 3a-d. This is also happening in PHATE representations (fig 5a-b). The objective would be to be closer to the representations obtained with the "original data" (subsampling data but before the "simulation of low efficiency or insertion of zeros"). Please include the plots for reference data (before downsampling) and downsampled data without imputation for all the rest of experiments.

- Previous two points concludes that imputation results do not show an actual improvement (not only SERM, but also any other imputation method). In fact, results for SERM are in agreement with performing almost no imputation. Maybe the best imputation is to perform no imputation?
- Moreover, we can not extract any conclusion for the application of this imputation method in a real scenario with actual original data (without filtering and subsetting cells and genes).

- There is only one point where results seem to improve using SERM: the evaluation of clustering, where authors include results for no imputation (figure 2b) or for the original data (figure 6). However, they only report these values in simulated data, and in the large datasets (cell atlas). In any case, they must provide accuracy of clustering for both "control" datasets in all figures: results for original data (what we would like to recover), and results for "downsampled data" (the starting point of imputation methods). In these way we can check if imputation is improving or not.

Regarding figures: please specify the percentage of "dropout" (a range of values were tested in the work) corresponding to the ones used to obtain the plots displayed in figures.

MS#: NCOMMS-22-04921A

Title: SERM: a self-consistent deep learning strategy for rapid and accurate gene expression recovery

The authors wish to thank the referees for their constructive comments. The manuscript has been revised to address the questions raised by the referees, as detailed below.

Reviewer #1 (Remarks to the Author):

1. I still have questions for the ROI step in the algorithm. In the revision, the authors explained that “ ... this issue is addressed by adding an optional step, in which the positions of cells or genes can be randomized through column/row permutations to ensure unbiased distribution in different parts of the matrix.” While I understand this might be true if all of the cells are collected from the same batch, I am not sure how this performs when the cells are collected from different batches. For the latter case, how do you permute the cells? Do you need to do any batch effects removal before doing the permutations?

Response: Thank you for bringing this important issue up. We have added the following paragraph in this revision:

“SERM on data from different batches: If a dataset consists of data from different batches, data from any of the batches can be used as reference. It should be noted that setting one batch-data as reference is a common practice in batch-effect correction techniques¹⁻³. Indeed, for the examples presented in supplementary Section 11, we found that the results of SERM imputation change little when a different batch is used as reference. The data distribution in SERM is learned from the reference batch and then applied to all batches for data imputation. In this case, the randomization of cells and genes as mentioned in the Discussion section is performed within their own batches before imputation calculation in SERM. After imputation, all the cells and genes are relocated to their original places. The SERM-imputed datasets from different batches are then integrated into a single dataset using a z-score operation for downstream analyses. Please see Fig. S43 for detailed workflow of SERM when a dataset contains data from different batches.”

Please see demonstration Python code in our Code Ocean capsule to test the usage of SERM on imputing the data with batch-effect.

Fig. S43. Workflow for batch-effect correction using SERM. Here, we show batch 1 as the reference batch. After the distribution learning, SERM imputes all the batches using the distribution. We note that any of the batches can be selected as the reference and there is little change in the performance of SERM when a different reference dataset is used.

2. Regarding my comment on batch effects, the authors responded that “Although batch effects are not considered explicitly in the formulation of SERM, the technique is generally less susceptible to data obtained by using different technologies or protocols (see Fig. S28, in which results from different data acquisition techniques with batch effects are shown).”, but Fig. S28 is not addressing my comment on batch effects. This figure only shows the performance of different imputation methods on two different datasets. I would like to see the authors appropriately evaluate the performance of SERM when batch effects are present.

Response: In this resubmission, we have added detailed benchmarking of SERM using UMAP visualizations and five quantitative indices on datasets with batch effect. We followed Refs. ^{1,4,5} for the benchmarking procedure. We took the single cell data from five different technologies (each denoting a different batch), combined them into one dataset and then used different imputation techniques to perform imputation. Following Refs. ^{1,4,5}, the imputation performance of different techniques is compared using UMAP visualizations, adjusted Rand, Average Silhouette width for batch and class, integration local inverse Simpson’s Index (LISI), and normalized mutual information indices. Please see Figs. S44-S46 for the results.

Fig. S46. Comparison of different imputation methods for batch effect correction on the human pancreas dataset. Bar plots of metrics AR, NMI, ASW_celltype, 1-ASW_batch, and iLISI for the 8 methods, together with the raw data (no imputation) for comparison. A higher value of AR, NMI, and ASW_celltype indicates better performance in clustering. A higher value of 1-ASW_batch and iLISI indicates better performance in batch mixing.

3. Regarding my comment on batch effects, the authors further responded that “Nonetheless, we would suggest using SERM on the data from different batches separately and then employing a dedicated batch-effect correction technique (such as Seurat 1).” The advantage of SERM is substantially reduced if it has to rely on a separate program such as Seurat to deal with batch effects. Most users would prefer to use a single program to conduct data analysis. Having to use the combination of SERM + another program (e.g. Seurat) is somewhat cumbersome for users.

Response: Data imputation and batch-effect removal are generally two separate problems in single-cell data analysis. Techniques like Seurat³, online iNMF⁶, and Harmony⁵ are developed dedicatedly for solving the later problem and cannot solve the first one. On the other hand, most imputation methods (MAGIC⁷, SAVER⁸, mclmpute⁹) are designed to solve the first problem only. However, the reviewer is right that it would be ideal if one method can perform both. With extensive benchmarking, we showed that SERM performs better than all the imputation techniques when the batch effect is present in the data (please see Figs. S44-S46 for the results).

Reviewer #2 (Remarks to the Author):

The reviewers have addressed all my concerns. I do not have further comments.

Response: We thank Reviewer 2 for his/her time and effort in reviewing the manuscript as well as constructive feedback.

Reviewer #3 (Remarks to the Author):

Authors change their strategy by using downsampled data instead of the zero-filled strategy to generate dropout-affected data to evaluate their imputation method (and others). It is not clear if this new "downsampled data" is really valid for evaluation, and for supporting their claims.

Response: We apologize that the process to create the downsampled data was not clear. In this resubmission, we have made it clear that we used the model in Ref. ⁸ (Huang et al, Nature Methods 15, 539-542, 2018) to create the downsampled and observed data. We also revised the terminology (reference data, observed data, downsampling, etc.) by strictly following Ref. ⁸ to avoid any confusion. To prove the versatility of SERM in single cell imputation, we have added a figure (Fig. S51) showing the performance of SERM for three different methods of creating dropout-affected data (Refs. ^{8,10,11}). It is seen that, in all cases, SERM can improve the data quality (which no other method can).

- The process to generate the dropout-affected data is not explained. It is a critical point since all conclusions derived from the use of data in this way.

Response: We have added the methodological details to generate the dropout-affected data. Please see subsection "Generating reference and observed datasets" in the methods section.

- It is not clear what are the "Original data" and the "Downsampled data" they use in their experiments and figures. It is not even clear if they perform imputation over the downsampled data and compare the results with the original data (which seems to be the case).

Response: We revised the terminology (reference data, observed data, downsampling, etc.) by strictly following Ref. ⁸ to avoid any confusion. We have now made it clear that Reference data is created by choosing the most informative 50-60% cells and 10-20% genes. Observed data is created from the reference data by simulating efficiency loss that introduces zeros (Please see methods section and Ref. ⁸). We imputed the observed data using SERM and other imputation methods and compared the imputed data with the reference data.

- Downsampled data is usually referred to extract a portion of data, which in this case would be a subset of cells and a subset of genes.

Response: In this revision, we have revised the terminology and used “Observed data” to represent the dropout-affected data following Ref. ⁸.

- Using this definition I do not see how imputation accuracy can be evaluated.

Response: Please see the above response, revised method section, supplementary Table 4 (added below) and Ref. ⁸ for details of the evaluation process of imputation accuracy. We have also added reference data and 7 dropout-affected datasets with different dropout rates to our Code-Ocean capsule so that the readers/reviewers can check the datasets.

Table 4. Number of cells and genes of analyzed datasets

Dataset	Raw	Reference	Observed
Cellular taxonomy	27,998 genes and 23,092 cells	2,422 genes and 12,162 cells	2,422 genes and 12,162 cells
Mammalian brain	30,341 genes and 18,194 cells	2,344 genes and 10,360 cells	2,344 genes and 10,360 cells
Mouse intestinal epithelium	15,971 genes and 7,216 cells	4,072 cells and 1,776 genes	4,072 cells and 1,776 genes
3D neural tissue data	22,567 genes and 4,280 cells	2,735 genes and 2,364 cells	2,735 genes and 2,364 cells
Zebrafish development	17,239 genes and 38,731 cells	2,341 genes and 20,014 cells	2,341 genes and 20,014 cells
EB differentiation	17,580 genes and 16,825 cells	2,282 genes and 9,754 cells	2,282 genes and 9,754 cells

- If authors mean a different concept, please use different terms and provide detailed information about the construction of these "downsampled datasets" and how it allows a proper evaluation of imputation methods.

Response: Thank you for the valuable suggestion. In this revision, we follow the nomenclature as Ref. ⁸ to avoid any confusion. We have added the following in the discussion section:

“In the evaluation of SERM, the reference data created by choosing only the cells and genes with high expressions provides a dropout-free representation. The observed data is created from the reference data by simulating efficiency loss that introduces zeros and represents the dropout-affected data. Different dropout rates were simulated by varying the parameters of Gamma distribution. The applications of SERM on a large number of datasets with different dropout rates shows its broad applicability.”

I needed to go deep into the references and code the authors provided trying to understand all these things:

- References and explanation of dropout-affected is not explained. Two references are provided within the text (18,37) and at least one of them is wrong (37).

[18. Miao, Z., Li, J. & Zhang, X. scRecover: Discriminating true and false zeros in single-cell RNA-seq data for imputation. 665323. URL <https://www.biorxiv.org/content/10.1101/665323v1>. DOI 10.1101/665323.]

[37. Abdelaal, T. et al. A comparison of automatic cell identification methods for single-cell RNA sequencing data. 20, 194. URL <https://doi.org/10.1186/s13059-019-1795-z>. DOI 10.1186/s13059-019-1795-z.]

Response: We have removed these references and only referred to Ref. ⁸ to avoid any confusion. We have added details on the process of creating the reference and dropout-affected data in the revised submission.

In response to reviewers they include 2 references [4,5] (4 could be right this time):

[4. Huang, M. et al. SAVER: Gene expression recovery for single-cell RNA sequencing. Nature Methods 15, 539–542 (2018).]

[5. Miao, Z., Li, J. & Zhang, X. scRecover: Discriminating true and false zeros in single-cell RNA-seq data for imputation. 665323 <https://www.biorxiv.org/content/10.1101/665323v1> (2019) doi:10.1101/665323.]

Response: We have removed Ref. 5 and only referred to Ref. ⁸ (i.e., [4] in the above).

In demo software provided by authors (<https://codeocean.com/capsule/0864915>) they only referenced Miao et al. for the generation of downsampled data, specifying the use of "countsSampling" function. This function basically reduces the number of counts in the sample. This may introduce some "simulated zeros" when the reduction is high, because of the integer nature of counts. If that is simply the way to create the downsampled data, it would have been very easy to explain and detail within the paper... (In both references there are other steps that may or may not be part of the analytical pipeline in this paper).

Response: We have revised the terminology (reference data, observed data, downsampling, etc.) by strictly following Ref. ⁸ to avoid any confusion. We have now made it clear that Reference data is created by choosing the most informative 50-60% cells and 10-20% genes as was done in Ref. ⁸. Observed data is created from the reference data by simulating efficiency loss that introduces zeros (Please see methods section and Ref. ⁸).

Note: this paper (Miao et al.) is not published in a peer-review journal, and it is just a preprint. The methodology used could be questionable.

Response: We have removed the paper by Miao et al..

- In Huang et al. authors generate "Reference datasets" by keeping only a portion of genes and cells (i.e. downsampled data) that they selected as high quality (genes where low amount of cells have zero expression, and cells with enough library size). From this reference data, they generated their "observed data" by simulating efficiency loss that "introduces zeros". They use these "observed data" as input for imputation, and compare results with the reference (downsampled) dataset.

Response: In the revised manuscript, we have clarified the issue by following the procedure and nomenclature as Ref. ⁸. We also use these "observed data" as input for imputation, and compare results with the reference dataset.

By checking the codes and toy data of the paper under review:

- "Original data" used by authors during the paper, seem to be something like "reference data" in huang et al. The example provided with the code (cellular taxonomy of the bone marrow stroma) only has 1093 genes and 23092 cells, so cells and genes seem to be filtered from the actual original data (with much more cells and genes). None of this preprocessing, filtering, etc. is named or explained in the paper and it is critical. Moreover, the term "original data" is confusing if it refers to processed data or a subset of data. Please, add number of cells/genes for each dataset in their corresponding places in Methods section.

- "Downsampled data" used by authors during the paper, seem to be something like "observed data" in Huang et al. The example provided with the code (cellular taxonomy of the bone marrow stroma) shows a very similar count matrix with respect to the "original" (reference), but with lower values.

- Authors seem to impute "observed data" (their "downsampled data") and compare the results with the "reference data" (their "original data"). It needs to be clear in the manuscript. Next points of this review are based on that assumption.

Response: Thank you for your careful review and valuable suggestions. We revised the nomenclature (reference data, observed data, downsampling, etc.) by strictly following Ref. ⁸ to avoid any confusion. We have now made it clear that Reference data is created by choosing the most informative 50-60% cells and 10-20% genes as suggested by Ref. ⁸. Observed data is created from the reference data by simulating efficiency loss that introduces zeros (Please see methods section and Ref. ⁸).

We have added details on the number of cells/genes for each dataset in the corresponding places in the methods section.

- I have serious doubts about the validity of their "downsampled data" to evaluate imputation. Based on the example they provide with the code (<https://codeocean.com/capsule/0864915>, cellular taxonomy of the bone marrow stroma):

- The percentage of zeros is very similar in reference (51.8%) and observed (downsampled to 1%, used for imputation) data (52.47%).

Response: We have added the percentage of zeros of the reference and observed data in the revised supplementary (see Fig. S47-added below). It is seen that for 10% sampling efficiency, the percentage of zeros in the reference and observed data is similar (around 63% for

cellular taxonomy dataset). However, for 0.1% efficiency, the percentages of zeros are very different (63% and 86.5%). Thus, the datasets of different sampling efficiencies describe the datasets affected by different rates of dropout.

- The Pearson correlation between the reference and downsampled data, prior to imputation, is already very high (0.89 for all genes together, and mean=0.85 median=0.87 when computing correlations for each gene independently). It is not clear from figure 4 first row, but those values seem to be higher than the ones obtained by any of the imputation methods tested (when comparing at 1%).

In any case, correlations of unimputed data need to be added in these plots (Figure 4, Figure 5c, Figure 6,...).

Response: We have added correlations of observed (unimputed) data with the reference data for all the figures. It is seen that, although the correlation of the dropout-affected observed data and reference data is very high (around 0.90 for cellular taxonomy data) for 10% efficiency, it is not the case for 0.1% (the correlation is only 0.45). For all dropout rates, SERM is able to improve the imputation accuracy (see Figs. 2-6 and S3, S4, S10).

- TSNE/UMAP plots do not reflect an improvement when performing imputation. Results with SERM are almost identical to the ones obtained with the "downsampled data" (no imputation). See figures 2a, 3a-d. This is also happening in PHATE representations (fig 5a-b). The objective would be to be closer to the representations obtained with the "original data" (subsampling data but before the "simulation of low efficiency or insertion of zeros"). Please include the plots for reference data (before downsampling) and downsampled data without imputation for all the rest of experiments.

Response: We have added results for "reference data", "observed data", and imputed data to clearly show the efficacy of SERM to discover the true clusters or patterns present in the reference data. Please see revised Figs. 2-6 for the revised TSNE/UMAP and PHATE plots. We have also added arrows to show the cluster or trajectory improvement in SERM results.

- Previous two points concludes that imputation results do not show an actual improvement (not only SERM, but also any other imputation method). In fact, results for SERM are in agreement with performing almost no imputation. Maybe the best imputation is to perform no imputation?

Response: In the revised submission, we have added visualization results for 0.1% sampling efficiency, where the superiority of SERM in comparison to the observed (unimputed) data is more obvious. However, it is true in many cases (especially when the sampling efficiency is higher than 5%), it is better to do "no imputation" than imputation by many methods except SERM (please see Fig. S3, S4, S10 for such results). Here, we emphasize that SERM improves the imputation for all cases (both low and high dropouts), which is made clear in the quantitative indices shown in Figs. 2-6, and S3 (added below), S4, S10.

- Moreover, we can not extract any conclusion for the application of this imputation method in a real scenario with actual original data (without filtering and subsetting cells and genes).

Response: In the analysis of human and mouse cell atlas datasets, no filtering and subsetting cells and genes were used and it was found that SERM is able to improve the clustering results (please see Figs.S35, S36, S39). In the revised manuscript, we have added another dataset (supplementary section 13, Fig. S49) (without filtering and subsetting cells and genes) and used different methods to recover the actual clusters. It is found that SERM performs substantially better than all other imputation techniques (see results below).

- There is only one point where results seem to improve using SERM: the evaluation of clustering, where authors include results for no imputation (figure 2b) or for the original data (figure 6). However, they only report these values in simulated data, and in the large datasets (cell atlas). In any case, they must provide accuracy of clustering for both "control" datasets in all figures: results for original data (what we would like to recover), and results for "downsampled data" (the starting point of imputation methods). In these way we can check if imputation is improving or not.

Response: We have added results for the reference data and observed data in the revised manuscript. We have added Figs. S5-S8 where we show the clustering results for all the datasets. Please see Figs. S3, S4, S10, where the percent improvement of imputation accuracy with reference to the observed data clearly shows that SERM improves the imputation accuracy in all datasets with all dropout rates.

Regarding figures: please specify the percentage of "dropout" (a range of values were tested in the work) corresponding to the ones used to obtain the plots displayed in figures.

Response: We have added the percentage of dropout to obtain the plots in the figure captions.

Reference:

1. Xu, X. *et al.* Propensity score matching enables batch-effect-corrected imputation in single-cell RNA-seq analysis. *Briefings in Bioinformatics* **23**, bbac275 (2022).
2. Loza, M., Teraguchi, S., Standley, D. M. & Diez, D. Unbiased integration of single cell transcriptome replicates. *NAR Genomics and Bioinformatics* **4**, lqac022 (2022).
3. Stuart, T. *et al.* Comprehensive Integration of Single-Cell Data. *Cell* **177**, 1888-1902.e21 (2019).
4. Tran, H. T. N. *et al.* A benchmark of batch-effect correction methods for single-cell RNA sequencing data. *Genome Biology* **21**, 12 (2020).
5. Korsunsky, I. *et al.* Fast, sensitive and accurate integration of single-cell data with Harmony. *Nat Methods* **16**, 1289–1296 (2019).
6. Gao, C. *et al.* Iterative single-cell multi-omic integration using online learning. *Nat Biotechnol* **39**, 1000–1007 (2021).
7. van Dijk, D. *et al.* Recovering Gene Interactions from Single-Cell Data Using Data Diffusion. *Cell* **174**, 716-729.e27 (2018).
8. Huang, M. *et al.* SAVER: Gene expression recovery for single-cell RNA sequencing. *Nature Methods* **15**, 539–542 (2018).
9. Mongia, A., Sengupta, D. & Majumdar, A. McImpute: Matrix Completion Based Imputation for Single Cell RNA-seq Data. *Front. Genet.* **10**, (2019).
10. Chen, M. & Zhou, X. VIPER: Variability-preserving imputation for accurate gene expression recovery in single-cell RNA sequencing studies. *Genome Biology* **19**, 196 (2018).
11. Tang, W. *et al.* bayNorm: Bayesian gene expression recovery, imputation and normalization for single-cell RNA-sequencing data. *Bioinformatics* **36**, 1174–1181 (2020).

Reviewer #1 (Remarks to the Author):

I have one remaining comment about batch effects & imputation. The authors said that "Data imputation and batch-effect removal are generally two separate problems in single-cell data analysis." This is not true. There are methods, e.g. scVI (PMCID: PMC6289068) and CarDEC (PMCID: PMC8494213), that are specifically designed to do both batch effects removal and imputation. These methods need to be acknowledged.

Reviewer #3 (Remarks to the Author):

Authors have addressed most of my concerns. However, there is still one important point to clarify/correct.

Major points:

- Figure 4. When comparing the last version of this figure (previous review) and the current one (which includes the results for data before imputation, which was requested by this reviewer, and is supposed to be the only difference between these versions), results of the Pearson coefficients for the same datasets and the same set of sampling efficiencies are completely different. Even if authors decided to rerun the entire evaluation from scratch (generating the observed data again), these differences are not acceptable if the algorithms involved are the same. In the previous version of the figure, correlation coefs systematically increase when the sampling efficiency is reduced (which makes sense, since low sampling efficiency increases the number of zeros/dropouts and makes imputation more difficult). This does not happen in the new version of this figure, where we can see many opposite tendencies (counter-intuitive) and also a huge number of negative correlations that did not exist before. Even the results of their own algorithm are quite different in some datasets; for instance mammalian brain (2nd row) and neural tissue (4th row).

Authors should check their codes in case they have some bugs when performing these evaluations.

- Figure 6. Same problem explained for Figure 4.

Minor points:

Generating reference and observed datasets  It should go after the introduction of the datasets ("Datasets" section). Also, authors could find a more concise way to describe or represent the (important information of) filters used in each dataset to generate the reference.

Fig 2b  To ease interpretation, Observed & Reference results should be the first or last items in the plots, and maybe using a different color than results obtained after imputation with different methods.

MS#: NCOMMS-22-04921B

Title: **SERM: a self-consistent deep learning strategy for rapid and accurate gene expression recovery**

The authors wish to thank the referees for their constructive comments. The manuscript has been revised to address the questions raised by the referees, as detailed below.

Reviewer #1 (Remarks to the Author):

I have one remaining comment about batch effects & imputation. The authors said that “Data imputation and batch-effect removal are generally two separate problems in single-cell data analysis.” This is not true. There are methods, e.g. scVI (PMCID: PMC6289068) and CarDEC (PMCID: PMC8494213), that are specifically designed to do both batch effects removal and imputation. These methods need to be acknowledged.

Response: Thank you for pointing this out. The sentence you mentioned appeared only in our point-to-point response (not in the manuscript). We have edited the discussion section and acknowledged the methods referred by you as follows:

“Batch-effect correction plays an important role in the scRNA-seq data analysis pipeline. A few methods that can perform both batch-effect correction and data imputation in the same calculation process, such as scVI ¹ and CarDEC ², have been reported. Although batch effects are not considered explicitly in the formulation of SERM, we note that the SERM is generally less susceptible to data obtained by using different technologies or protocols (see supplementary section 11 for benchmarking of SERM on datasets with batch-effect). The reason behind this is that SERM enforces a learned distribution from a single set of data (e.g., a batch of data) to all the data from different batches. Even if the data of different batches have different distributions, SERM forces them to follow the learned distribution, alleviating any potential bias and artifacts (see Methods section).”

Reviewer #3 (Remarks to the Author):

Authors have addressed most of my concerns. However, there is still one important point to clarify/correct.

Major points:

- Figure 4. When comparing the last version of this figure (previous review) and the current one (which includes the results for data before imputation, which was requested by this reviewer, and is supposed to be the only difference between these versions), results of the Pearson coefficients for the same datasets and the same set of sampling efficiencies are completely different. Even if authors decided to rerun the entire evaluation from scratch (generating the observed data again), these differences are not acceptable if the algorithms involved are the same. In the previous version of the figure, correlation coeffs systematically increase when the sampling efficiency is reduced (which makes sense, since low sampling efficiency increases the number of zeros/dropouts and makes imputation more difficult). This does not happen in the new version of this figure, where we can see many opposite tendencies (counter-intuitive) and also a huge number of negative correlations that did not

exist before. Even the results of their own algorithm are quite different in some datasets; for instance mammalian brain (2nd row) and neural tissue (4th row).

Authors should check their codes in case they have some bugs when performing these evaluations.

Response: We thank you for your careful reading of the manuscript and for the comment. The changes in Fig. 4 (and Fig. 6) as noted by you arise from the change of reference data in the last (i.e., the 2nd) resubmission. In the 1st resubmission, we chose all the cells with selected highly

variable genes to create the reference data. Based on your comments, we followed Ref.³ to create the reference data by selecting subsets of high-quality cells and genes in the last submission. Please see our uploaded cellular taxonomy data in the last two resubmissions. We note that the performance variation of SERM for different runs is negligible. The reviewer/reader can test this by running our Code-Ocean capsule multiple times.

Previously, a custom code written by us was used to compute the Pearson correlation values, which became problematic in dealing with irregular numbers such as nan and inf. The problem has been fixed in this resubmission – we used standard functions from NumPy package: `numpy.nan_to_num` to replace the irregular numbers and `numpy.corrcoef` in the calculations of the Pearson correlation values for Figures 4 and 6. It is seen that as the sampling efficiency increases (reduction of dropout), the performance of most methods improves. We have added the following sentences in the results section of the revised manuscript:

“In Fig. 6 (and Fig. 4), it is seen that as the sampling efficiency increases (reduction of dropout), the performance of most methods improves. We noticed that the performance of the DeepImpute is, in some cases, better than other methods in the first two sampling efficiencies (0.1% and 0.2%) with comparable results to the SERM. Overall, Magic and SAUCIE are less accurate in genomic data imputation when compared to other techniques.”

- Figure 6. Same problem explained for Figure 4.

Response: We have corrected Figures 4 and 6 as discussed above.

Minor points:

Generating reference and observed datasets  It should go after the introduction of the datasets ("Datasets" section). Also, authors could find a more concise way to describe or represent the (important information of) filters used in each dataset to generate the reference.

Response: We have moved the “Generating reference and observed datasets” after the “Datasets” section.

Based on your suggestion, we have concisely described the filtering process in the main manuscript and added details of the filtering for each dataset in supplementary section 12.

Fig 2b  To ease interpretation, Observed & Reference results should be the first or last items in the plots, and maybe using a different color than results obtained after imputation with different methods.

Response: Thank you for the suggestion. We have used different colors for the results of the observed and reference datasets in the revised Fig. 2b to ease interpretation.

Reference:

1. Lopez, R., Regier, J., Cole, M. B., Jordan, M. I. & Yosef, N. Deep generative modeling for single-cell transcriptomics. *Nat. Methods* **15**, 1053–1058 (2018).
2. Lakkis, J. *et al.* A joint deep learning model enables simultaneous batch effect correction, denoising and clustering in single-cell transcriptomics. *Genome Res.* gr.271874.120 (2021) doi:10.1101/gr.271874.120.
3. Huang, M. *et al.* SAVER: Gene expression recovery for single-cell RNA sequencing. *Nat. Methods* **15**, 539–542 (2018).

Reviewer #3 (Remarks to the Author):

Authors have addressed all my concerns

MS#: NCOMMS-22-04921C

Title: **SERM: a self-consistent deep learning strategy for rapid and accurate gene expression recovery**

The authors wish to thank the referees for their constructive comments. Below we add the reviewer's comment and our response.

Reviewer #3 (Remarks to the Author): Authors have addressed all my concerns

Response: We want to thank the reviewer for his/her time and effort in reviewing our manuscript.